# Designing Time Series Experiments in A/B Testing with Transformer Reinforcement Learning

**Xiangkun Wu**[1,*]
School of Mathematical Sciences
Zhejiang University
Hangzhou, China
12235031@zju.edu.cn

**Qianglin Wen**[2,*]
Yunnan Key Laboratory of Statistical Modeling and Data Analysis
Yunnan University
Kunming, China
qianglin@mail.ynu.edu.cn

**Yingying Zhang**[3,*]
School of Statistics
East China Normal University
Shanghai, China
yyzhang@fem.ecnu.edu.cn

**Hongtu Zhu**[4]
University of North Carolina at Chapel Hill
Chapel Hill, NC, USA
htzhu@email.unc.edu

**Ting Li**[5,†]
School of Statistics and Data Science
Shanghai University of Finance and Economics
Shanghai, China
tingli@mail.shufe.edu.cn

**Chengchun Shi**[6,†]
Department of Statistics
London School of Economics and Political Science
London, UK
C.Shi7@lse.ac.uk

[*] Equal contribution.  [†] Corresponding authors.

## Abstract

A/B testing has become a gold standard for modern technological companies to conduct policy evaluation. Yet, its application to time series experiments, where policies are sequentially assigned over time, remains challenging. Existing designs suffer from two limitations: (i) they do not fully leverage the entire history for treatment allocation; (ii) they rely on strong assumptions to approximate the objective function (e.g., the mean squared error of the estimated treatment effect) for optimizing the design. We first establish an impossibility theorem showing that failure to condition on the full history leads to suboptimal designs, due to the dynamic dependencies in time series experiments. To address both limitations simultaneously, we next propose a transformer reinforcement learning (RL) approach which leverages transformers to condition allocation on the entire history and employs RL to directly optimize the MSE without relying on restrictive assumptions. Empirical evaluations on synthetic data, a publicly available dispatch simulator, and a real-world ridesharing dataset demonstrate that our proposal consistently outperforms existing designs.

## 1 Introduction

This paper studies A/B testing, which has been deployed by numerous technological companies including multi-sided platforms such as Airbnb, DoorDash and Meituan, major e-commerce firms such

as Amazon and Alibaba, and leading companies in search and social networking such as Google, Meta and LinkedIn. The primary goal of A/B testing is policy evaluation: comparing a newly developed web design, policy or product (referred to as the *treatment*) against an existing version (referred to as the *control*). At its core, A/B testing applies causal inference methodologies (see e.g., Imbens & Rubin, 2015; Ding, 2024; Wager, 2024) to design online experiments, randomize experimental units to treatment and control groups, and infer the average treatment effect from the data collected.

We focus on applications where experimental units are exposed to sequences of treatments over time, resulting in data that form a time series (see e.g., Bojinov & Shephard, 2019; Farias et al., 2022; Shi et al., 2023b; Li et al., 2024a; Xiong et al., 2024b). One motivating application is A/B testing in large-scale ridesharing platforms such as Uber, Lyft, and Didi Chuxing. These companies operate as two-sided marketplaces that match passengers with drivers to offer efficient and convenient transportation services. They frequently develop and update various policies for order dispatch (Xu et al., 2018; Wan et al., 2021; Zhou et al., 2021), vehicle repositioning (Pouls et al., 2020; Jiao et al., 2021), pricing and subsidies (Fang et al., 2017; Chen et al., 2019), and employ A/B testing for validation (Chamandy, 2016; Shi et al., 2023a). These policies are implemented sequentially over time. Moreover, in ridesharing, the number of passengers requesting rides (demand) and the drivers' online time (supply) evolve dynamically over time (see e.g., Luo et al., 2024, Figure 1). Together with the sequence of policies, these variables form three interacting time series: policies influence both demand and supply, while demand and supply also interact with each other.

A/B testing in such time series experiments faces three critical challenges:

1. **Carryover effects** are ubiquitous in these experiments (Shi et al., 2023b; Xiong et al., 2024b; Ni & Bojinov, 2025; Wen et al., 2025). That is, policies often exhibit *delayed* effects: a policy implemented at each time can affect both the company's current outcome and their future outcomes, leading to the violation of the classical stable unit treatment value assumption (see e.g., Imbens & Rubin, 2015). In ridesharing, policies such as order dispatch, vehicle repositioning, pricing, and subsidies all have delayed effects. They alter the spatial distribution of drivers across a city, and such changes occur gradually, resulting in the delay in their impact on future outcomes. Ignoring these carryover effects and applying classical A/B testing methods by treating the time series data as independent and identically distributed often leads to insignificant average treatment effect (ATE) estimator (Shi et al., 2023b).

2. **Small treatment effects** make it extremely challenging to distinguish between new and existing policies in terms of their impact on the company's outcomes (Athey et al., 2023). In ridesharing, improvements from newly developed order dispatch policies are very modest, typically ranging from only 0.5% to 2% (Tang et al., 2019).

3. **Limited duration** of the experiments further hinders accurate estimation of the ATE. Specifically, time series experiments are often restricted to a few weeks (Bojinov et al., 2023), resulting in relatively small sample sizes for A/B testing.

To address the last two challenges, this paper investigates how to carefully design time series experiments to improve the accuracy of the estimated ATE, while accounting for carryover effects to address the first challenge. The design of experiments is a classical problem in statistics (Fisher, 1935), motivated by applications in agriculture, and has recently gained growing attention in machine learning (see Section 2). Its objective is to optimally generate experimental data to maximize the information obtained for accurate estimation. In our context of A/B testing, this translates to deciding, at each time, which policy (treatment or control) to implement so that the ATE estimated from the resulting experimental data achieves the smallest possible mean squared error (MSE).

Our primary contribution lies in the development of such a design for A/B testing, leveraging modern neural network architectures and advanced machine learning algorithms (see Figure 1 for a summary of our proposal):

• Theoretically, we present a negative result showing that existing designs, which restrict treatment allocation to depend only on the initial action, the current observation or a short history (see Figure 2 for a graphical illustration), can be suboptimal. Specifically, we establish an **impossibility theorem** to prove that when restricting to the class of doubly robust estimators (Tsiatis, 2007) for ATE estimation, it is generally *impossible* for allocation strategies that do not fully leverage the entire history to achieve optimality, at least asymptotically.

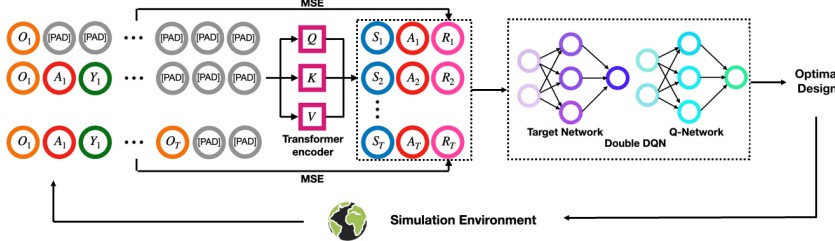

Figure 1: Illustration of the proposed transformer reinforcement learning algorithm. Our algorithm employs a transformer encoder to summarize the full historical context and to produce the state $\{S_t\}_t$. These states are then fed into a double deep Q-network agent, which outputs an optimal policy that minimizes the mean squared error of the ATE estimator (encoded as the (negative) reward $\{R_t\}_t$).

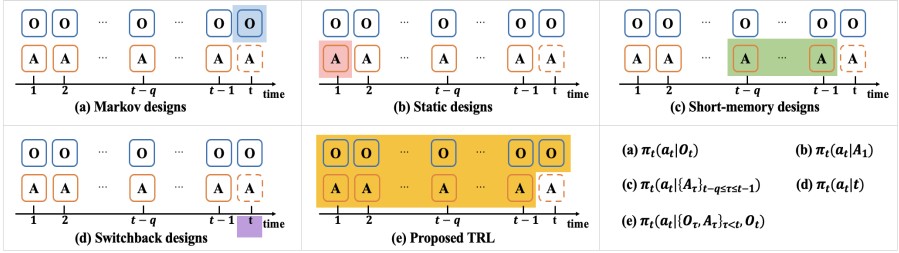

Figure 2: Graphical illustrations of treatment allocations for existing designs (a)–(d) and our proposed design (e). Specifically, existing designs condition treatment allocation on only the current observation (a), the initial action (b), a limited history (c), or the time index (d). In contrast, the proposed design conditions on the entire historical information (e).

- Building on this insight, we employ **transformers** (Vaswani et al., 2017) to allow the determination of policies at each time to depend on the full data history, in order to capture the dynamic dependencies inherent in time series experiments.

- We next notice that existing works often rely on strong assumptions (see Section 2.2 for details) to approximate the MSE as the objective function for design optimization. To address this limitation, we employ **reinforcement learning** (RL, Sutton & Barto, 2018) to directly minimize the MSE, without requiring these assumptions.

- Empirically, we demonstrate the usefulness of the proposed design using (i) synthetic data, (ii) a publicly available dispatch simulator that realistically mimics the behaviors of drivers and passengers, and (iii) a real dataset collected from a ridesharing company.

In summary, our proposal can handle carryover effects and temporal dependencies, requires minimal assumptions on the data generating process and attains substantial empirical gains, all of which being achieved by harnessing modern computational power.

## 2 RELATED WORKS

Our proposal is related to three branches of research: A/B testing, experimental designs and RL for optimization. We review related works in each branch in detail below.

### 2.1 A/B TESTING

The literature on A/B testing is extensive; we refer readers to Larsen et al. (2024) and Quin et al. (2024) for two recent reviews. Traditional A/B testing methods estimate treatment effects using the difference-in-means estimator between treatment and control groups across various business metrics such as the gross merchandise value, to determine whether the new policy outperforms the existing

one. However, as noted in the introduction, this approach overlooks delayed effects, yielding biased estimates in the presence of carryover effects.

There are four main approaches to handling carryover effects:

1. The first applies importance sampling (see e.g., Precup et al., 2000; Zhang et al., 2013; Thomas et al., 2015; Guo et al., 2017; Bojinov & Shephard, 2019; Hu & Wager, 2023; Zhou et al., 2025), which reweights the outcome at each time step using the ratio of the target policy to the behavior policy that generated the experimental data. However, this estimator often suffers from high variance – a phenomenon known as the curse of horizon (Liu et al., 2018; Xie et al., 2019).

2. The second builds on the standard difference-in-means estimator but corrects its bias by discarding a subset of data after switching between treatment and control (Hu & Wager, 2022). This approach, however, reduces the effective sample size, which can be particularly problematic given the limited duration of experiments.

3. The third adopts an RL framework by modeling the experimental data as a Markov decision process (MDP, Puterman, 2014), which enables the application of existing off-policy evaluation algorithms (e.g., Le et al., 2019; Hao et al., 2021; Shi et al., 2021b; 2022; Chen & Qi, 2022; Uehara et al., 2022; Bian et al., 2025) for A/B testing; see e.g., Glynn et al. (2020); Farias et al. (2022); Tang et al. (2022); Farias et al. (2023); Shi et al. (2023b; 2024); Wen et al. (2025). This approach, however, requires the time series to satisfy the Markov assumption, which can be violated in practice (Sun et al., 2024).

4. The last relaxes the Markov assumption by using partially observable MDPs (POMDPs), particle filters or more general time series models for A/B testing (see e.g., Menchetti et al., 2021; Sun et al., 2024; Liang & Recht, 2025; Ni & Bojinov, 2025). Nonetheless, these approaches still rely on specific model assumptions (e.g., linearity, parametric structure, or short-term dependence), which limits their ability to capture the complex, nonlinear, and long-range temporal dependencies that frequently arise in practice.

Most of the aforementioned works focus on estimating the ATE given experimental data, whereas we address a different question: how to design and generate the experimental data itself. While both the third approach and ours leverage RL, their objectives differ: the third uses RL as a modeling framework for ATE estimation under carryover effects, whereas we use RL as a computational tool for MSE minimization.

To address other challenges such as small treatment effects and limited experimental duration, many algorithms have been proposed for sample efficient A/B testing. Beyond experimental design, other approaches include developing more accurate ATE estimators (Chen & Simchi-Levi, 2023; Chen et al., 2024; Sakhi et al., 2025), more powerful hypothesis tests (Wang et al., 2025; Zhang et al., 2025b), and borrowing information from historical or concurrent experimental data (Li et al., 2023b; Jung & Bellot, 2024; Li et al., 2024b; Wu et al., 2025).

Finally, we note that there are two additional lines of research: one focuses on sequential monitoring, which studies how to conduct A/B testing across multiple interim stages while controlling the overall type-I error or false discoveries (Johari et al., 2017; Yang et al., 2017; Shi et al., 2021a; Maharaj et al., 2023; Wan et al., 2023; Waudby-Smith et al., 2024; Lindon & Kallus, 2025), and the other on handling interference effects beyond carryover effects, such as spillover effects across multiple experimental units (Munro et al., 2021; Li et al., 2022; Li & Wager, 2022; Leung & Loupos, 2022; Shi et al., 2023a; Dai et al., 2024; Lu et al., 2024; Zhan et al., 2024).

## 2.2 Experimental designs

The design of experiments has been extensively studied across various disciplines, driven by a wide range of applications from biology, medicine, psychology and engineering. In the following, we structure the related literature into three categories: (i) early contributions in statistics, primarily motivated by applications in clinical trials; (ii) recent advances in machine learning, particularly those related to RL; and (iii) works in management science, operations research and other related fields, with a focus on modern A/B testing in technology industries:

(i) Early works in statistics have introduced several optimality criteria (e.g., D-optimality and A-optimality), and optimal designs like Neyman allocation as well as their generalizations, to op-

timize the covariate distribution (see e.g., Neyman, 1934; de Aguiar et al., 1995; Jones & Goos, 2009; Yin & Zhou, 2017; Loux, 2013; Jones et al., 2021; Li & Del Castillo, 2024; Atkinson, 2025, and the references therein). Our proposal is related to those on optimizing treatment allocation strategies in clinical trials, where a variety of designs have been developed, including covariate-adaptive design (Lin et al., 2015; Zhou et al., 2018; Bertsimas et al., 2019; Ma et al., 2024), response-adaptive design (Hu & Rosenberger, 2006; van der Laan, 2008; Robertson et al., 2023; Wei et al., 2023; 2025), and covariate-adjusted response-adaptive design (Zhang et al., 2007; Hu et al., 2015; Lin et al., 2015; Zhao et al., 2022; Li et al., 2024c). However, most of these designs do not account for carryover effects. An exception is crossover designs (Hedayat & Stufken, 2003; Li et al., 2015; Louis et al., 2019; Koner, 2024). But they typically require long washout periods, which reduce the effective sample size, limiting their practicality for modern A/B testing with small treatment effects and limited durations.

(ii) More recently, the design of experiments has attracted growing attention in machine learning (see, e.g., Kato et al., 2024; Liu et al., 2024a; Weltz et al., 2024; Zhang & Wang, 2024; Pillai et al., 2025; Zhu et al., 2025). In particular, a line of research leverages deep learning and RL to numerically compute optimal designs (see, e.g., Foster et al., 2021; Blau et al., 2022; Lim et al., 2022; Annadani et al., 2024; Ai et al., 2025; Barlas & Salako, 2025). Our approach shares the same underlying principle with these works but differs in two aspects. First, while these works consider generic design problems, we focus specifically on minimizing the MSE of the ATE estimator in A/B testing. Second, we propose to leverage the transformer neural network architecture to allow the treatment allocation strategy to condition on the entire history, for more effective designs under temporal dependencies (see Section 3 for our rationale for using transformers). In the RL literature, the design problem is also referred to as behavior policy search (Hanna et al., 2017; Mukherjee et al., 2022; Liu & Zhang, 2024; Liu et al., 2025). Additionally, classical D-optimal designs have been employed for more efficient policy learning in MDPs (Agarwal et al., 2019, Section 3.3).

(iii) There is a growing line of works on experimental design for A/B testing in technological industries, much of which appears in management science and operations research (see e.g., Ugander et al., 2013; Viviano, 2020; Bajari et al., 2021; Wan et al., 2022; Bajari et al., 2023; Johari et al., 2022; Leung, 2022; Jia et al., 2023; Ni et al., 2023; Viviano et al., 2023; Liu et al., 2024b; Xiong et al., 2024a; Zhao, 2024a;b; Missault et al., 2025; Xiong, 2025; Yu et al., 2025; Zhang et al., 2025a). Our proposal is particularly related to those that focus on time series experiments (Glynn et al., 2020; Hu & Wager, 2022; Basse et al., 2023; Bojinov et al., 2023; Chen & Simchi-Levi, 2023; Li et al., 2023a; Sun et al., 2024; Xiong et al., 2024b; Ni & Bojinov, 2025; Wen et al., 2025). However, these works suffer from two limitations. First, as demonstrated by our impossibility theorem (Theorem 1), their failure to condition on the full history for treatment allocation can be suboptimal. Second, they require strong assumptions to simplify the optimization of the MSE, such as (a) assuming an MDP model (Glynn et al., 2020; Hu & Wager, 2022; Li et al., 2023a; Wen et al., 2025), (b) imposing certain time series model assumptions (Xiong et al., 2024b; Sun et al., 2024; Ni & Bojinov, 2025), (c) assuming delayed effects persist for at most a few time periods (Basse et al., 2023; Bojinov et al., 2023; Chen & Simchi-Levi, 2023). In contrast, our proposal does not require these assumptions.

## 2.3 RL FOR OPTIMIZATION

RL is a modern machine learning paradigm for learning optimal policies to maximize expected cumulative rewards in sequential decision making. Owing to its effectiveness in policy optimization, RL has been adopted as a computational tool to tackle complex combinatorial optimization problems, including the classical traveling salesman and knapsack problems (Bello et al., 2016), the vehicle routing, orienteering, and prize-collecting traveling salesman problems (Kool et al., 2018), the device placement problem (Mirhoseini et al., 2017; 2018) and dag constraints (Zhu et al., 2019); see the survey by Mazyavkina et al. (2021) for a comprehensive review. These works parameterize the solution as an optimal policy, and set the rewards to the objective function, which enables the application of RL for optimization. Building on this principle, our proposal applies RL to the specific problem of minimizing the MSE of the ATE estimator, while integrating the transformer neural network architecture to effectively capture the temporal dependencies inherent in time series experiments.

## 3 TRANSFORMER RL FOR EXPERIMENTAL DESIGN

**Problem setup**. Consider an online experiment conducted by a technology company, which lasts for $T$ non-overlapping time intervals. At the beginning of the $t$th time interval, the decision maker observes a set of features denoted by $O_t$. They then determine whether to implement the control policy $(-1)$ or the new policy $(1)$ at $t$th time interval, represented by an action $A_t \in \{-1, 1\}$. At the end of the interval, a numerical outcome $Y_t \in \mathbb{R}$ is observed, where larger values indicate better outcomes. For example, on a ridesharing platform, $O_t$ may include market features to characterize demand and supply prior to the $t$th interval; $A_t$ may represent order dispatch, repositioning, or subsidy policies; and $Y_t$ can be the percentage of completed or accepted orders (completion or response rate) or driver income. Notice that in time series experiments, data are temporally dependent: the outcome $Y_t$ and future observation $O_{t+1}$ at each time may depend not only on the current observation-action pair, but also on past data history. We denote their conditional distribution, given the data history, by $\mathcal{P}_t$, i.e.,

$$\mathcal{P}_t(\mathcal{Y}, \mathcal{O} \mid H_{t-1}, O_t, A_t) = \mathbb{P}(Y_t \in \mathcal{Y}, O_{t+1} \in \mathcal{O} \mid H_{t-1}, O_t, A_t),$$

where $\mathcal{Y}$ and $\mathcal{O}$ denote the subsets of outcome and observation spaces, respectively, and $H_{t-1} = \{O_1, A_1, Y_1, \ldots, O_{t-1}, A_{t-1}, Y_{t-1}\}$ denotes the entire history up to time $t - 1$.

After the experiment concludes, the decision maker collects the observation-action-outcome triplets to estimate the *ATE*, defined as

$$\text{ATE} = \frac{1}{T} \sum_{t=1}^{T} \left[ \mathbb{E}_1(Y_t) - \mathbb{E}_{-1}(Y_t) \right], \tag{1}$$

where $\mathbb{E}_1$ and $\mathbb{E}_{-1}$ denote expectations under hypothetical scenarios in which all actions are set to the new or control policy, respectively. This estimator is then used to determine which policy (whether new or control) to deploy. As reviewed in Section 2.1, there is a large literature on ATE *estimation* given the experimental data.

In contrast, this paper studies a complementary *design* question: *given an ATE estimation procedure, how shall we assign actions over time during the experiment so as to minimize the MSE of the resulting ATE estimator?* Mathematically, let $\pi_t$ denote the treatment allocation strategy used to generate $A_t$ given the past data history, we aim to determine the optimal $\pi = \{\pi_t\}_t$ that minimizes

$$\text{MSE}(\pi) = \mathbb{E}_\pi(\widehat{\text{ATE}} - \text{ATE})^2, \tag{2}$$

where $\widehat{\text{ATE}}$ denotes the ATE estimator and $\mathbb{E}_\pi$ indicates that the expectation is taken over data generated under $\pi$.

**Limitations of existing designs**. With these formulations in place, we are now prepared to present our impossibility theorem. To analytically calculate the MSE, we need to specify the ATE estimator. Here, we restrict our attention to the class of doubly robust estimators, due to wide use in policy evaluation in both bandit settings and sequential decision making with and without the Markov assumption (Jiang & Li, 2016; Chernozhukov et al., 2018; Kallus & Uehara, 2022; Liao et al., 2022). Under mild regularity assumptions (Chernozhukov et al., 2018; Kallus & Uehara, 2022), these estimators are asymptotically unbiased, so their MSE is equivalent to their asymptotic variance (denoted by $\text{Var}(\pi)$), i.e.,

$$T\text{MSE}(\pi) = T\text{Var}(\pi) + o(1),$$

which attains the semiparametric efficiency bound – the smallest achievable MSE within a large class of regular and asymptotically linear estimators (e.g., Tsiatis, 2007).

**Theorem 1** (Impossibility theorem). *Suppose we set $\widehat{ATE}$ to the double robust estimator. Then there exist data generating processes $\{\mathcal{P}_t\}_t$ under which the optimal policy $\pi$ that minimizes $\text{Var}(\pi)$ depends on the entire past history for all $1 \leq t \leq T$, and this optimal policy is unique.*

In other words, it is impossible for a treatment allocation strategy in which $\pi_t$ omits dependence on any observation, action, or outcome at any time step to be optimal.

As commented earlier, existing designs suffer from two limitations: (i) Their treatment allocation strategy $\pi$ does not rely on the full history, which can be suboptimal according to Theorem 1. (ii) They often impose restrictive assumptions on the data generating process to simplify the calculation of the MSE. We provide several examples to elaborate:

- *Markov designs*: Glynn et al. (2020) employ a finite MDP to model the experimental data. They consider Markov designs in which each $\pi_t$ depends on the past history only through the current observation $O_t$ and develop an approach to learn the optimal Markov design by solving a succinct convex optimization problem.

- *Static designs*: Li et al. (2023a) extend the classical Neyman allocation strategy to MDPs. Their optimal design is static: within each day, the same action is assigned throughout time. Consequently, $\pi_t$ depends only on the first action.

- *Short-memory designs*: Sun et al. (2024) adopt the classical ARMA($p, q$) model – a special partially observable MDP (POMDP) – to handle non-Markov environments. They prove that under a small signal assumption, the optimal $\pi_t$ depends only on the past $q$ actions. Meanwhile, Ni & Bojinov (2025) restrict $\pi_t$ to depend only on the most recent action.

- *Switchback designs*: Bojinov et al. (2023) and Wen et al. (2025) study switchback designs, which alternate between the control and new policy at fixed time intervals. Bojinov et al. (2023) show that the optimal switch duration depends crucially on the number of periods that carryover effects persist. Wen et al. (2025) allow carryover effects to persist throughout the entire experiment by adopting an MDP model and show that the optimal switch duration depends on reward autocorrelations and the magnitude of carryover effects. In both cases, $\pi_t$ depends only on the time index $t$ (and the initial action for Wen et al., 2025).

In summary, these strategies rely on limited historical information – such as the first action, the current observation, the time index, or a short history – and impose MDP or ARMA model assumptions, or constraints on carryover effects.

**Our proposal**. To address both limitations simultaneously, we propose a transformer reinforcement learning (TRL) algorithm for optimally designing online experiments. Our approach integrates two key components: (i) an RL framework that directly optimizes the MSE without relying on restrictive modeling assumptions, and (ii) a transformer-based neural network architecture that enables treatment allocation to condition on the entire history.

**(i) An RL framework**. We assume access to a simulation environment that can generate $(O_t, A_t, Y_t)_t$ that approximates the experimental data generating process (we will discuss how to construct such an environment later). We then apply RL to interact with this environment to optimize the design. To apply RL, we define the state-action-reward triplet as follows:

- *State*: We define the state $S_t$ as the full history $\{O_1, A_1, Y_1, \ldots, O_{t-1}, A_{t-1}, Y_{t-1}, O_t\}$ up to time point $t$, in order to capture all potential temporal dependencies that might influence the optimal treatment allocation at that time;

- *Action*: The action is the same to $A_t$, determining which policy (standard or new) to implement at time $t$;

- *Reward*: The reward at each time $t$ is set to

$$R_t = -\alpha^{T-t} \big[\widehat{\text{ATE}}(t) - \text{ATE}_{mc}\big]^2, \tag{3}$$

where $\widehat{\text{ATE}}(t)$ denotes the ATE estimator computed based on all data triplets up to time $t$, $\text{ATE}_{mc}$ denotes the Monte Carlo estimator learned from the simulator, and $\alpha \in [0, 1)$ denotes a discount factor.

We make a few remarks on the reward. Note that $R_t$ differs from $Y_t$; it serves as a proxy for the underlying MSE, rather than the company's actual outcome. Specifically, before running RL, we use Monte Carlo to generate a large number of trajectories following either the standard or the new policy, and compute $\text{ATE}_{mc}$, which is then treated as the ground truth to approximate the oracle ATE. During RL, at each time $t$, we compute $\widehat{\text{ATE}}(t)$ using the observation-action-outcome triplets collected up to time $t$ and measure its squared deviation from $\text{ATE}_{mc}$ as an approximation of the MSE. This squared error is then discounted by $\alpha^{T-t}$ to downweight earlier steps whose ATE estimators are less accurate due to smaller sample sizes. In the extreme case where $\alpha = 0$, only the last reward $R_T$ matters. Finally, the negative sign ensures that smaller MSEs yield larger rewards.

In our collaboration with the ridesharing company, physical simulators built upon historical data are available to simulate the effects of various policies. These simulators can be readily used as

the simulation environment for RL. Indeed, before running online experiments, most policies are evaluated offline using such simulators to demonstrate their effects. If no simulator is available, the experiment can be conducted sequentially: after each day, we use the collected experimental data to estimate the data generating process and construct a simulator, then apply the proposed procedure to design the experiment for the next day. This process is repeated where we update both the simulator and the design as more data become available. Furthermore, we can combine both experimental data and additional synthetic data from the simulator to estimate the ATE (see e.g., Lin et al., 2024; Xiong, 2025). Our procedure is fairly general, as the ATE estimator $\widehat{\text{ATE}}(t)$ can be computed using any algorithm, including such data integration approaches.

**(ii) Transformer-based DDQN**. We develop a variant of double deep Q-network (Van Hasselt et al., 2016) that leverages transformer architectures to learn the optimal policy. Specifically, we define the Q-function as

$$Q_t(S_t, A_t) = \mathbb{E}_{\pi^{opt}} \left[ \sum_{k=t}^{T} \gamma^{k-t} R_{k-t} \,\middle|\, S_t, A_t \right],$$

where $\pi^{opt}$ denotes the optimal policy, which is our target treatment allocation strategy and is greedy with respect to the Q-function. We propose to use transformers with masked self-attention to parameterize the Q-function. There are two advantages to this parameterization: (i) The dimension of the Q-function's input $S_t$ varies over time, and transformers can readily handle such varying-length inputs to encode the entire history; (ii) Compared to other recurrent neural networks, transformers are able to capture long-term dependencies (Zeng et al., 2023; Shi & Shide, 2025), ensuring that all relevant historical variables that may contribute to the optimal policy are effectively utilized.

We repeatedly sample trajectories from the simulation environment to learn the Q-function. The loss function is defined as the squared loss between the Q-function and the learning target. Optimization is performed using AdamW with cosine learning-rate scheduling, gradient clipping, and mixed-precision training for improved efficiency. A summary of our procedure is illustrated in Figure 1.

## 4 EXPERIMENTS

In this section, we conduct numerical experiments to evaluate the finite-sample performance of our proposal. Comparisons is made among the following types of designs:

- **TRL**: the proposed transformer RL design.
- **TMDP/NMDP**: Designs proposed by Li et al. (2023a) tailored for MDPs. These designs switch treatments on a daily basis and assign the same treatment within each day.
- **Switchback designs** proposed by Hu and Wager (**HW**), Bojinov, Simchi-Levi, and Zhao (**BSZ**), Xiong, Chin, and Taylor (**XCT**), and Wen, Shi, Yang, Tang, and Zhu (**WSY**). These designs switch treatments every few time intervals. WSY uses fixed switching intervals, whereas HW, BSZ, and XCT consider regular switchback designs with random switching intervals. The difference among HW, BSZ, and XCT lies in the ATE estimators they employ. All these designs involve certain hyperparameters, for which we consider their optimal configurations.

We evaluate the aforementioned designs in three simulation environments: (1) a synthetic simulation environment; (2) a real-data-based simulation environment and (3) a publicly available dispatch simulator. Details of these environments are provided in Appendix A.2. For each environment, we compute the MSE of the ATE estimators using data generated from each design over 400 simulation replications.

**Synthetic simulator**. We consider a simulation environment where data are i.i.d. across days. Each day is divided into $M$ non-overlapping time intervals, during which a two-dimensional feature $O_t$ evolves according to a linear MDP, and the outcome $Y_t$ is generated with a linear reward function. We vary the number of days $n \in \{30, 35, 40, 45\}$ and set the number of intervals $M = 4$. We examine Settings (i)–(iv), varying both variances and transition structures. The transition coefficients are stationary over time and are specified in Appendix A.2. Figure 3 reports the empirical MSEs of various ATE estimators under different designs for Settings (i)-(ii), along with their confidence intervals (CIs). We make three observations:

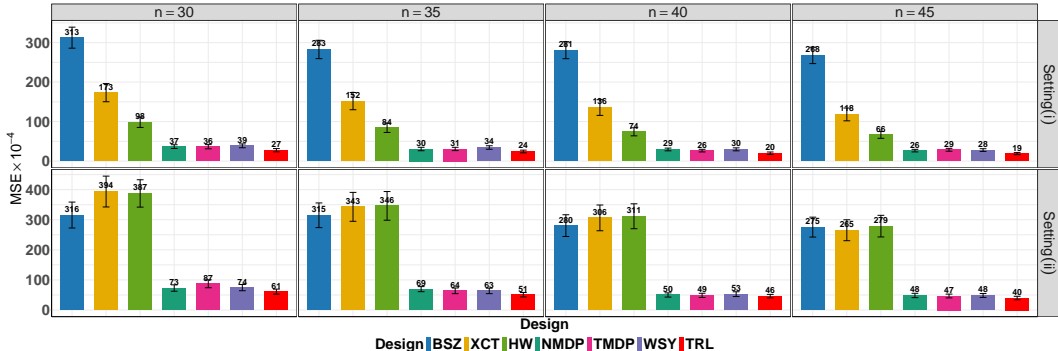

Figure 3: Barplots of empirical MSEs under different designs with their confidence intervals in the synthetic environment, across varying variances (Setting (i)) and transition structures (Setting (ii)).

(a) The proposed TRL generally outperforms all other baselines, achieving the smallest MSE and shortest CI in most settings.

(b) Switchback designs such as XCT, HW and BSZ perform poorly in this environment, suffering from significantly larger MSEs than the rest of the designs.

(c) NMDP and TMDP perform comparable to TRL, with slightly larger MSEs. This is expected, as these designs are optimized for settings with i.i.d. trajectories across days, and each day indeed follows an MDP. Despite their theoretical optimality, TRL still performs better, likely due to its transformer architecture, which effectively leverages historical information and yields superior performance in finite samples.

Additional experimental results for Settings (iii)-(iv) are presented in Figure 6 in the Appendix, showing similar patterns.

**Real data-based simulator**. We use an A/A dataset from a world-leading ridesharing company to construct the simulator. The A/A experiment spans 40 days during which a single order dispatch policy is employed throughout. Within each day, the trajectory data are divided into $M = 12$ or 24 non-overlapping time intervals. The observation vector is two-dimensional, consisting of (i) the number of order requests and (ii) the total online time of drivers in the previous interval, capturing the demand and supply of the two-sided marketplace. The reward is defined as the total driver income earned within each interval. However, the raw dataset cannot be directly used to evaluate different designs. To address this, we first estimate the data generating process from the A/A dataset and then construct the simulation environment using the wild bootstrap (Wu et al., 1986). We consider a range of settings where the number of days $n \in \{21, 28, 35, 42\}$, the number of intervals $M \in \{12, 24\}$, and the policy improvement (measured by the ratio of the ATE to the average outcome under the control policy) is selected from $\{2.5\%, 5\%\}$, reflecting the effects typically observed for newly developed dispatch policies (Tang et al., 2019).

Figure 4 reports the empirical MSEs for various designs, together with their confidence intervals, under a policy improvement of $5\%$. As before, the proposed TRL design consistently outperforms all other designs, particularly when $n$ is small. Interestingly, TMDP and NMDP yield significantly larger MSEs in this environment. This is due to the positive correlation observed in outcomes in the A/A dataset: in such settings, designs that switch treatments on a daily basis such as TMDP and NMDP are no longer optimal (Xiong et al., 2024b; Wen et al., 2025). Figure 7 in the Appendix reports results under a $2.5\%$ policy improvement, showing similar patterns.

**Public dispatch simulator**. Following Xu et al. (2018) and Li et al. (2023a), we employ a publicly available physical dispatch simulator[1] that generates realistic patterns of passenger and driver behavior in a ridesharing marketplace to evaluate different designs. In this simulator, drivers and orders operate on a $9 \times 9$ spatial grid over 20 time steps per day, and orders are canceled if they remain unserved for an extended period. We compare the MDP-based order dispatch policy of Xu et al. (2018) with a distance-based method that minimizes the total driver–passenger distance during

---

[1]https://github.com/callmespring/MDPOD.

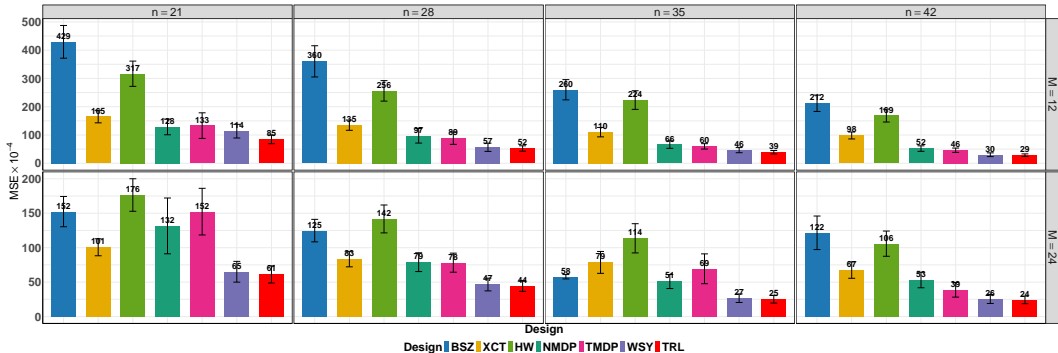

Figure 4: Barplots of the empirical MSEs under different designs in the real-data-based simulation, with a $5\%$ performance improvement from the new policy.

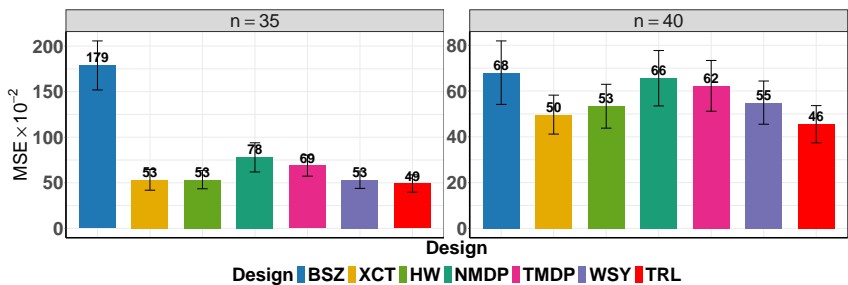

Figure 5: Barplots of empirical MSEs under different designs with their confidence intervals in the dispatch environment, across different days.

dispatch. The outcome of interest is the total revenue at each time step, and the observation variables again include both supply and demand. Simulations are conducted over $n \in \{35, 40\}$ days, with each design evaluated using 100 orders and a fixed fleet of 25 drivers.

Figure 5 reports the empirical MSEs, demonstrating that the proposed TRL consistently outperforms all baselines. As $n$ increases, TRL achieves lower mean MSEs. While switchback designs mitigate spatial and temporal carryover effects by alternating treatments and thereby reducing bias—leading to smaller MSEs than NMDP and TMDP—the proposed TRL attains the lowest MSEs.

## 5 CONCLUSION

This paper studies the design of time series experiments in A/B testing. We propose a novel transformer RL approach to enable treatment allocation to depend on the entire data history and relax restrictive modeling assumptions required by existing designs. Future work could extend our framework beyond time series experiments to more general experimental settings involving multiple experimental units, where spillover effects between units may arise.

## REPRODUCIBILITY STATEMENT

We have taken extensive steps to ensure the reproducibility of our results. For experiments based on publicly available datasets, we will release all source code, including data preprocessing, model training, and evaluation scripts, in the supplementary materials.

Due to data use agreements, certain real-world datasets cannot be shared publicly. Nevertheless, we provide a detailed description of their characteristics in Appendix A.2 together with comprehensive instructions for the associated simulators, and a visual empirical analysis of real-world datasets in Appendix B.8.

To mitigate randomness and allow for an objective evaluation, we performed 400 independent runs and reported the mean squared error (MSE) together with its confidence interval (CI).

Experiments were conducted on NVIDIA A100 (40GB) and RTX 2080 GPUs, with the fastest synthetic experiment training within about three GPU hours. In addition, we provide scripts to reproduce all figures in the main paper for experiments conducted on public or synthetic data.

## ETHICS STATEMENT

This work does not raise any known ethical concerns. The datasets used are either publicly available or synthetically generated, and do not contain personally identifiable information. Our methods are intended for advancing research in experimental design and causal inference, and we do not foresee direct misuse. Nevertheless, as with any machine learning research, there is a potential risk of unintended applications. We encourage future users of our code and methods to apply them responsibly and in accordance with ethical guidelines.

## ACKNOWLEDGMENT

We thank the anonymous referees and the meta-reviewer for their constructive comments, which have significantly improved this manuscript. Xiangkun Wu's research was supported by the National Key Research and Development Program of China (Grant No. 2024YFC2511003). Qianglin Wen's research was supported by the National Key R&D Program of China (Grant No. 2022YFA1003701). Yingying Zhang's research was supported by the National Natural Science Foundation of China (Grant No. 12471280). Ting Li's research was supported by the National Natural Science Foundation of China (Grant No. 12571304), the Shanghai Pujiang Program (Grant No. 24PJIC030), the CCF–DiDi GAIA Collaborative Research Funds, and the Program for Innovative Research Team of Shanghai University of Finance and Economics.

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

## LARGE LANGUAGE MODEL (LLM) USE DECLARATION

We used **GPT-5 Pro** as an auxiliary tool in the preparation of this paper. Its usage was limited to:

- **Writing support**: grammar correction, sentence restructuring, and improving readability of the manuscript (approximately 5–8% of the final text).
- **Code optimization**: suggestions for improving Python simulation scripts (e.g., tensor operations, parallelization on GPUs, and visualization enhancements, approximately 10–15% of the whole codes).
- **Technical clarity**: checking alternative phrasings of definitions and equations for better presentation.

LLMs were **not** used for:

- generating novel research ideas,
- creating mathematical proofs,
- designing experiments, or
- interpreting results.

All LLM-generated suggestions were carefully reviewed, verified, and substantially revised by the authors. The final responsibility for all text, code, and conclusions lies solely with the authors.

## A APPENDIX

### A.1 PROOF OF THEOREM 1

*Proof.* Let $H_{t-1}$ denote the set of observation-action-outcome triplets up to time $t - 1$. Consider the data generating distributions $\{\mathcal{P}_t\}_t$ that satisfy the following three conditions:

**Condition 1** (Conditional mean independence assumption (CMIA)). The conditional mean of $Y_t$ given $O_t$, $A_t$ is independent of $H_{t-1}$, for any $t$.

**Condition 2** (Conditional independence assumption (CIA)). The conditional distribution of $O_{t+1}$ given $H_t$ depends only on $\{O_j\}_{j \leq t}$, for any $t$.

**Condition 3** (History-dependent conditional variance ratio (HCVR)). Let $\sigma_t^2(H_t, O_t, A_t)$ denote the conditional variance of $Y_t$ given $O_t$, $A_t$ and $H_t$. Then, for each $t$, $\sigma_t(H_t, O_t, A_t)$ is positive almost surely, and the ratio

$$\frac{\sigma_t(H_t, O_t, 1)}{\sigma_t(H_t, O_t, -1)}$$

depends on all variables in $H_t$ and $O_t$. That is, no subset of these variables is sufficient to fully recover this ratio.

It is straightforward to construct data generating processes $\{\mathcal{P}_t\}_t$ to satisfy these assumptions.

Following Tsiatis (2007); Kallus & Uehara (2020), the double robust estimator under CMIA and CIA can be shown to take the following form:

$$\widehat{\text{ATE}} = \frac{1}{T} \sum_{t=1}^{T} \left[ \mu_t(O_t, 1) - \mu_t(O_t, -1) + \mathbb{I}(A_t = 1) \frac{Y_t - \mu_t(O_t, 1)}{\pi_t(1|H_t, O_t)} - \mathbb{I}(A_t = -1) \frac{Y_t - \mu_t(O_t, -1)}{\pi_t(-1|H_t, O_t)} \right],$$

where $\mu_t(o, a) = \mathbb{E}(Y_t | O_t = o, A_t = a)$. It follows from the CMIA that the residuals $Y_t - \mu_t(O_t, A_t)$ are uncorrelated so that

$$\text{Var}(\widehat{\text{ATE}}) = \frac{1}{T^2} \sum_{t_1=1}^{T} \sum_{t_2=1}^{T} \left[ \text{Cov} \left( \mu_{t_1}(O_{t_1}, 1) - \mu_{t_1}(-1, O_{t_1}), \mu_{t_2}(1, O_{t_2}) - \mu_{t_2}(O_{t_2}, -1) \right) \right]$$

$$+ \frac{1}{T^2} \sum_{t=1}^{T} \left[ \mathbb{E} \left( \mathbb{I}(A_t = 1) \frac{Y_t - \mu_t(O_t, 1)}{\pi_t(1|H_t, O_t)} \right)^2 + \mathbb{E} \left( \mathbb{I}(A_t = -1) \frac{Y_t - \mu_t(O_t, -1)}{\pi_t(-1|H_t, O_t)} \right)^2 \right]$$

$$= \frac{1}{T^2} \sum_{t_1=1}^{T} \sum_{t_2=1}^{T} \left[ \text{Cov} \left( \mu_{t_1}(O_{t_1}, 1) - \mu_{t_1}(O_{t_1}, -1), \mu_{t_2}(O_{t_2}, 1) - \mu_{t_2}(O_{t_2}, -1) \right) \right]$$

$$+ \frac{1}{T^2} \sum_{t_1=1}^{T} \left[ \mathbb{E}[\frac{\sigma^2(H_t, O_t, 1)}{\pi_t(1|H_t, O_t)}] + \mathbb{E}[\frac{\sigma^2(H_t, O_t, -1)}{\pi_t(-1|H_t, O_t)}] \right].$$

$$(4)$$

By minimizing the conditional variance of the above ATE estimator with respect to the assign probability, we can obtain the optimal allocation $\pi_t^{opt}$ at each time $t$, given by

$$\pi_t^{opt}(1|H_t, O_t) = \frac{\sigma_t(H_t, O_t, 1)}{\sigma_t(H_t, O_t, 1) + \sigma_t(H_t, O_t, -1)}.$$

When the right-hand-side depends on both $O_t$ and $H_t$, so does the optimal allocation strategy. Under HCVR, the optimal policy $\pi_t^{opt}$ depends on all variables in $O_t$ and $H_t$, for any $t$.

To complete the proof, we show the uniqueness of $\{\pi_t^{opt}\}_t$. Given its uniqueness, it is impossible for treatment allocation strategy in which $\pi_t$ omits dependence on any historical variable at any time step to be optimal. The proof is thus completed.

For any policy $\pi$, it follows from equation 4 that

$$\mathrm{Var}(\pi) - \mathrm{Var}(\pi^{opt}) = \frac{1}{T^2} \sum_{t=1}^{T} \mathbb{E}\big[\phi_t\big(\pi_t; H_t, O_t\big)\big], \tag{5}$$

where

$$\phi_t(\pi; h, o) := \frac{\sigma_t^2(h, o, 1)}{\pi} + \frac{\sigma_t^2(h, o, -1)}{1 - \pi}, \qquad 0 < \pi < 1.$$

With some calculations, we have

$$\phi_t\big(\pi_t^{\mathrm{opt}}; h, o\big) = \{\sigma_t(h, o, 1) + \sigma_t(h, o, -1)\}^2.$$

It follows that for any $0 < \pi < 1$,

$$\begin{aligned}
\Delta_t(\pi; h, o) &:= \phi_t(\pi; h, o) - \phi_t(\pi_t^{\mathrm{opt}}; h, o) \\
&= \frac{\big(\sigma_t(h, o, 1)(1 - \pi) - \sigma_t(h, o, -1)\,\pi\big)^2}{\pi(1 - \pi)} \geq 0.
\end{aligned} \tag{6}$$

Given the positiveness of $\sigma_t$ in Condition 3, the equality holds if and only if $\pi = \pi_t^{opt}$.

By equation 5 and equation 6, for any allocation rule $\pi$ we obtain

$$\mathrm{Var}\{\widehat{\mathrm{ATE}}(\pi)\} - \mathrm{Var}\{\widehat{\mathrm{ATE}}(\pi^{\mathrm{opt}})\} = \frac{1}{T^2} \sum_{t=1}^{T} \mathbb{E}\big[\Delta_t\big(\pi_t(1 \mid H_t, O_t); H_t, O_t\big)\big] \geq 0,$$

where the inequality holds if and only if $\pi_t = \pi_t^{opt}$, for any $t$. This completes the proof for the uniqueness of the optimal policy.

$\square$

## A.2 DETAILS OF SIMULATION EXPERIMENT.

In this section, we present the detailed state transition specifications of the simulation environments and report additional experimental results.

**Synthetic simulator (Continued).** At the beginning of each day $i \in \{1, \ldots, n\}$, the initial state $O_{i,1} \in \mathbb{R}^2$ is drawn with from the standard normal distribution $O_{i,1} \sim \mathcal{N}(0, I_2)$. Moreover, regardless of the experimental design method employed, the initial states $\{O_{i,1}\}_{i=1}^n$ are mutually independent across different days. For each day $i$ and each time step $m$, the outcome $Y_{i,m}$ and the next state $O_{i,m}$ evolve according to the following linear transition equations with Gaussian noise:

$$Y_{i,m} = \alpha_m + \beta_m^\top O_{i,m} + \gamma_m A_{i,m} + \varepsilon_{i,m}^Y, \qquad \varepsilon_{i,m}^Y \sim \mathcal{N}(0, \sigma_y^2), \quad m = 1, \ldots, M, \tag{7}$$

$$O_{i,m} = \phi_m + \Phi_m O_{i,m-1} + \Gamma_m A_{i,m-1} + \varepsilon_{i,m}^O, \quad \varepsilon_{i,m}^O \sim \mathcal{N}(0, \sigma_o^2 I_2), \quad m = 2, \ldots, M. \tag{8}$$

where, $\alpha_m \in \mathbb{R}$ denotes the intercept term, $\beta_m \in \mathbb{R}^2$ is the coefficient vector for the state $O_{i,m}$ in the outcome equation, and $\Phi_m \in \mathbb{R}^{2 \times 2}$ is the state transition matrix. Together with $\gamma_m \in \mathbb{R}$, $\phi_m \in \mathbb{R}^2$, and $\Gamma_m \in \mathbb{R}^2$, these parameters characterize the direct and indirect effects of the state and the action on both the outcome and the state transitions. The noise terms $\{\varepsilon_{i,m}^Y, \varepsilon_{i,m}^O\}$ are assumed to be mutually independent across both time $m$ and individuals $i$.

Following Luo et al. (2024), the true ATE can be defined as in (10). The generated panel data $\{O_{i,m}, A_{i,m}, Y_{i,m} : 1 \leq i \leq n, 1 \leq m \leq M\}$ can be flattened into a single time-series trajectory $\{O_t, A_t, Y_t\}t = 1^T$, where $t = (i - 1)M + m$. Together with our definitions of state and reward,

this yields $\{S_t, A_t, R_t\}t = 1^T$. A similar procedure is presented in the next subsection. We consider $n \in \{30, 35, 40, 45\}$ days and $M = 4$. We consider three settings.

Setting (i).

$$\alpha_m = 0, \qquad \beta_m = (0.6, 0.2)^\top, \qquad \gamma_m = 0.2, \qquad \text{for all } m = 1, \ldots, M,$$

and

$$\phi_m = \begin{bmatrix} 0 \\ 0 \end{bmatrix}, \qquad \Phi_m = \begin{bmatrix} 0.5 & 0.1 \\ 0.0 & 0.6 \end{bmatrix}, \qquad \Gamma_m = \begin{bmatrix} 0.1 \\ 0.05 \end{bmatrix}, \qquad \text{for all } m = 2, \ldots, M.$$

The standard deviations are set to be $\sigma_y = \sigma_o = 0.2$. (ii). The settings are similar to (i) except that

$$\Phi_m = \begin{bmatrix} 0.6 & 0.2 \\ 0.5 & 0.6 \end{bmatrix},$$

to examine how our methods perform under varying dynamics, and $\sigma_y = \sigma_o = 0.3$. (iii) The settings are similar to (i) except that $\sigma_y = \sigma_o = 0.3$. (iv) The settings are similar to those in (iii) except that $\beta_m = (0.3, 0.1)^\top$.

Figure 6 reports the MSEs under Settings (iii)-(iv). The results indicate that the proposed **TRL** generally achieves lower mean MSEs than the other methods.

---

**Algorithm 1** Bootstrap-based Simulator

---

1: **Input:** Real data $\{(O_{i,m}, Y_{i,m}) : 1 \le i \le n, \ 1 \le m \le M\}$; adjustment parameters $(\delta_1, \delta_2)$; Monte Carlo truth $\text{ATE}_{\text{mc}}$; bootstrap sample size $n$; random seed; number of bootstrap replications $B$.
2: **Output:** Synthetic time-series trajectories $\{(S_t^b, A_t^b, R_t^b)_{t=1}^T\}_{b=1}^B$ with $B$ replications.
3: Compute least-squares estimates $\{\widehat{\alpha}_m\}, \{\widehat{\beta}_m\}, \{\widehat{\phi}_m\}, \{\widehat{\Phi}_m\}$ in model equation 9, treatment-effect parameters $\{\widehat{\gamma}_m\}, \{\widehat{\Gamma}_m\}$, and residuals from equation 11.
4: **for** $b = 1$ **to** $B$ **do**
5:     Sample $n$ days from $\{1, \ldots, N\}$ with replacement and draw $\xi_i^b \sim \mathcal{N}(0, 1)$.
6:     Generate state $S_t^b$ and action $A_t^b$ at time $t$ using the transformer-based DDQN with an epsilon-greedy policy, where $t = (i - 1)M + m$.
7:     Generate pseudo outcomes $\{\widehat{Y}_{i,m}^b\}_{i,m}$ and observable states $\{\widehat{O}_{i,m}^b\}_{i,m}$ via:

$$\widehat{Y}_{i,m}^b = [1, \ (\widehat{O}_{i,m}^b)^\top, \ A_{i,m}^b] \begin{pmatrix} \widehat{\alpha}_m \\ \widehat{\beta}_m \\ \widehat{\gamma}_m \end{pmatrix} + \xi_i^b \, \widehat{e}_{i,m},$$

$$\widehat{O}_{i,m+1}^b = [\widehat{\phi}_m, \ \widehat{\Phi}_m, \ \widehat{\Gamma}_m] \begin{pmatrix} 1 \\ \widehat{O}_{i,m}^b \\ A_{i,m}^b \end{pmatrix} + \xi_i^b \, \widehat{E}_{i,m}.$$

8:     Compute $\{\text{ATE}_{\text{SB}}^b\}_b$ by OLS, and calculate the reward $R_t^b$ using equation 12.
9: **end for**

---

**Real data-based simulator (Continued).** We use a dataset spanning from May 17, 2019, to June 25, 2019, with one-hour intervals, resulting in $M = 24$ time units per day. In line with the bootstrap-based simulation procedure of Wen et al. (2025, Algorithm 5), we construct a simulator to generate synthetic experimental data. In particular, we apply linear models to both the reward function and the expected value of the next state, resulting in the following set of linearity assumptions:

$$\begin{cases} \mu_t(O_m, A_m) = \alpha_m + O_m^\top \beta_m + \gamma_m A_m, \\ \mathbb{E}(O_{m+1} | A_m, O_m) = \phi_m + \Phi_m O_m + \Gamma_m A_m, \end{cases} \tag{9}$$

where $\alpha_m$ and $\gamma_m$ are real-valued, $\beta_m, \phi_m$, and $\Gamma_m$ are vectors in $\mathbb{R}^d$, and $\Phi_m \in \mathbb{R}^{d \times d}$. Under Model equation 9, as outlined in Luo et al. (2024), the ATE can be expressed as

$$\frac{2}{M} \sum_{m=1}^M \gamma_m + \frac{2}{M} \sum_{m=2}^M \beta_m^\top \Big[ \sum_{k=1}^{m-1} (\Phi_{m-1} \Phi_{m-2} \ldots \Phi_{k+1}) \Gamma_k \Big], \tag{10}$$

where the product $\Phi_{m-1} \dots \Phi_{k+1}$ is treated as an identity matrix if $m - 1 < k + 1$. The first term on the right-hand side (RHS) of equation 10 represents the direct effect of actions on immediate rewards, while the latter term accounts for the delayed or carryover effects of previous actions. The equation 10 motivates the use of ordinary least squares (OLS) regression to obtain the relevant estimators, which are then substituted into equation 10 to compute the final ATE estimator. This yields the estimators $\{\widehat{\alpha}_m\}_m$, $\{\widehat{\beta}_m\}_m$, $\{\widehat{\phi}_m\}_m$ and $\{\widehat{\Phi}_m\}_m$. However, $\{\gamma_t\}_m$ and $\{\Gamma_m\}_t$ remain unidentifiable, since $A_m = 0$ almost surely. We then calculate the residuals in the reward and observable state regression models based on these estimators as follows:

$$\widehat{e}_{i,m} = Y_{i,m} - \widehat{\alpha}_m - O_{i,m}^\top \widehat{\beta}_m, \quad \widehat{E}_{i,m} = O_{i,m+1} - \widehat{\phi}_m - \widehat{\Phi}_t O_{i,m}. \tag{11}$$

To generate simulation data with varying sizes of treatment effect, we introduce the treatment effect ratio parameter $\lambda$ and manually set the treatment effect parameters $\widehat{\gamma}_m = \delta_1 \times (\sum_i Y_{i,m}/N)$ and $\widehat{\Gamma}_m = \delta_2 \times (\sum_i O_{i,m}/N)$. The treatment effect ratio essentially corresponds to the ratio of the ATE and the baseline policy's average return. The treatment effect ratio is chosen from $\{2.5\%, 5\%\}$.

Finally, to create a synthetic dataset spanning $n$ days, the synthetic dataset is flattened into a single time-series trajectory indexed by $t = 1, \dots, T$, with $T = nM$ and $t = (i-1)M + m$ for $i \in [n]$ and $m \in [M]$. State is established by full history up to time point $t$, and action is generated according to transformer-based DDQN with an epsilon-greedy policy. We then sample i.i.d. standard Gaussian noises $\{\xi_i\}_{i=1}^n$. For the $i$-th day, we uniformly sample an integer $I \in \{1, \dots, N\}$, set the initial observation to $O_{I,1}$, and generate rewards and states according to equation 9 with the estimated $\{\widehat{\alpha}_m\}_m$, $\{\widehat{\beta}_m\}_m$, $\{\widehat{\phi}_m\}_m$, $\{\widehat{\Phi}_m\}_m$, the specified $\{\widehat{\gamma}_m\}_m$ and $\{\widehat{\Gamma}_m\}_m$, and the error residuals given by $\{\xi_i \widehat{e}_{i,m} : 1 \le m \le M\}$ and $\{\xi_i \widehat{E}_{i,m} : 1 \le m \le M\}$, respectively. This ensures that the error covariance structure of the synthetic data closely resembles that of the real datasets. Based on the simulated data, we estimate the ATE at the end of each day using OLS. According to the definition in equation 3, the non-zero proxy reward is then calculated as

$$R_t = \begin{cases} -\alpha^{\,n-i} \big[\widehat{\text{ATE}}(t) - \text{ATE}_{\text{mc}}\big]^2, & \text{if } t = iM, \\ 0 & \text{otherwise,} \end{cases} \tag{12}$$

where the Monte Carlo truth $\text{ATE}_{\text{mc}}$ can be pre-estimated by generating large datasets under the control policy and the new policy, respectively. Up to the time horizon $T$, we collect the full trajectory $\{S_t, A_t, R_t\}_{t=1}^T$, which is then used to train our transformer-based DDQN. A summary of the bootstrap-assisted procedure is provided in Algorithm 1.

Figure 7 reports the MSEs for a true treatment effect ratio of 2.5%. The proposed TRL attains the lowest mean MSEs across all baselines, with mean MSEs decreasing as $n$ increases.

**Public dispatch simulator (Continued).** In this example, we provide a more detailed description of the environment ,similar to Xu et al. (2018). We examine the interactions between drivers and orders in a $9 \times 9$ spatial grid with a duration of 20 time steps. Drivers are constrained to move vertically or horizontally by only one grid at each time step, while orders can only be dispatched to drivers within a Manhattan distance of 2. An order will be canceled if not being assigned to any driver for a long time. The cancellation time follows a truncated Gaussian distribution with a mean of 2.5, and a standard deviation of 0.5, ranging from 0 to 3 on the temporal axis. To generate realistic traffic patterns that mimic a morning peak and a night peak, we model residential and working areas separately, and orders' starting locations are sampled using a two-component Gaussian mixture distribution. The locations are then truncated to integers within the spatiotemporal grid. Orders' destinations and drivers' initial locations are randomly sampled from a discrete uniform distribution on the grid. The parameters of the mixture of Gaussians are as follows.

$$\pi^{(1)} = 1/3, \pi^{(2)} = 2/3, \mu^{(1)} = [3, 3, 2], \mu^{(2)} = [6, 6, 14], \sigma^{(1)} = [2, 2, 2], \sigma^{(2)} = [2, 2, 2],$$

The three dimensions correspond to the spatial horizontal and vertical coordinates, and the temporal coordinate respectively.
Here, we treat the summary statistics of the system, namely the number of active orders and the number of active drivers at each time step, as a two-dimensional state variable $O$. By the number of active orders we mean the number of orders that satisfy feasibility constraints (e.g., not yet expired), and by the number of active drivers we mean the number of drivers who are idle and thus available to accept new orders. The action $A$ (dispatch decision) takes two values, 1 and $-1$, where $A = 1$ corresponds to the MDP-based policy and $A = -1$ corresponds to the distance-based policy. Specifically,

1 denotes the *MDP-based action*, while $-1$ denotes the *distance-based action*. The MDP-based action corresponds to a dispatching policy that selects orders so as to maximize the sum of the drivers' long-term expected revenue, taking into account the future benefits of current matches. In contrast, the distance-based action corresponds to choosing the assignment that minimizes the total pickup distance of all matches at time $m$. The outcome $Y_m$ is defined as the total revenue of all matched orders at time $m$. We first generate the information for each driver and each order, and then apply the Sinkhorn algorithm to obtain the optimal matching that maximizes the overall assignment objective. Based on these generated large-scale data, we learn the long-term value function $V$ associated with the action of each driver. Subsequently, we construct the MDP-based dispatching policy for order allocation using the learned value function $V$. Based on the learned MDP-based policy and the distance-based policy, we generate a large number of samples. Using these samples, we fit the state transition dynamics with a neural network model.

$$\hat{\varepsilon}_{i,m}^Y = Y_{i,m} - \hat{Y}_{i,m}, \quad \hat{Y}_{i,m} = \hat{f}_t(O_{i,m}, A_{i,m}), \quad m = 1, \ldots, M, \tag{13}$$

$$\hat{\varepsilon}_{i,m}^O = O_{i,m} - \hat{O}_{i,m}, \quad \hat{O}_{i,m} = \hat{g}_t(O_{i,m-1}, A_{i,m-1}), \quad m = 2, \ldots, M. \tag{14}$$

where $\hat{f}_m$ and $\hat{g}_m$ denote neural network predictors trained by minimizing the squared error loss. We compute the residuals at each time step across all samples and estimate their mean and variance. We then assume the residuals follow Gaussian distributions with the corresponding means and variances. Based on this, we construct a simulator in which the predicted values from the neural networks $\hat{f}_m$ and $\hat{g}_m$ are perturbed by Gaussian noise, with the outputs rounded and truncated to better mimic the real-world setting (where the number of active drivers and active orders are integers). Using this simulator, we generate 20,000 Monte Carlo samples under the all+1 and all−1 action sequences to obtain the ground truth ATE. For the experimental design methods, our TRL estimator employs the Least-Squares Temporal Difference (LSTD) approach to estimate the ATE. This differs from the first two experiments, since the present environment is not linear, and thus the more general LSTD method is required.

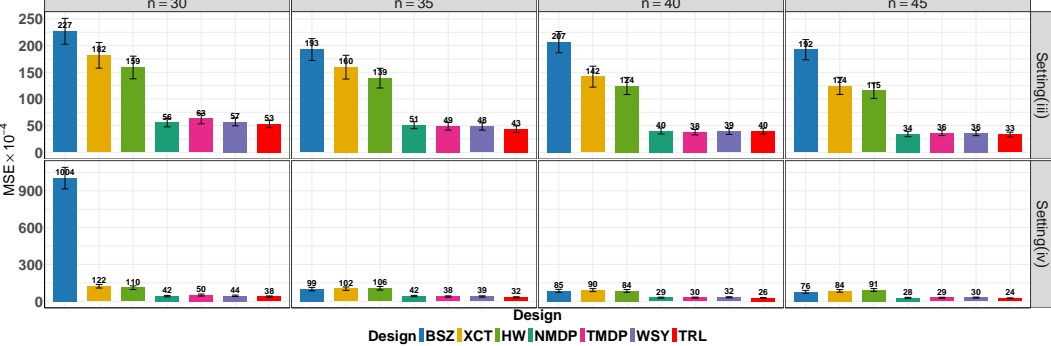

Figure 6: Barplots of empirical means of MSEs with $M = 4$ in Settings (iii) and (iv) of the **Synthetic simulator**.

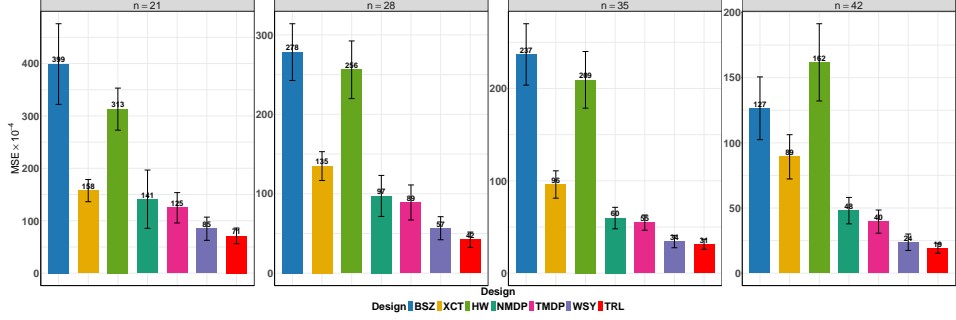

Figure 7: Barplots of the empirical MSEs under different designs in the real-data-based simulation with $M = 12$, with a $2.5\%$ performance improvement from the new policy.

Table 1: Empirical MSE between TRL, LSTM, and different attention window sizes.

| TRL | LSTM | Window size 6 | Window size 12 | Window size 24 |
|---|---|---|---|---|
| **0.002750** | 0.002975 | 0.003002 | 0.002915 | 0.02871 |

# B    ADDITIONAL EXPERIMENTS

In this section, we present additional experiments to further support our method and provide a more comprehensive empirical evaluation, including ablation studies, analyses of hyperparameter and simulator sensitivity and robustness, evaluations of generality and computational complexity, an assessment of simulator availability in practice, empirical correlation analysis based on real A/A data, and sensitivity experiments under varying levels of autocorrelation.

## B.1    ABLATION STUDY ON LSTM MODELS WITH DIFFERENT ATTENTION WINDOW SIZES

We conduct the experiment in Setting (i) to compare our full-history Transformer (TRL) against two key ablations: (1) an LSTM-based DDQN with an identical training budget, and (2) Transformer variants with restricted attention windows.

The results in Table 1 show that sequence modeling based on LSTM already offers a substantial advantage over traditional methods such as TMDP (MSE: 0.0036). Our full-history Transformer (TRL) further improves upon LSTM, achieving a lower MSE. Moreover, we observe that, for the Transformer variants, increasing the attention window size consistently leads to better performance, indicating the benefit of exploiting longer-range temporal dependencies.

## B.2    ABLATION STUDY ON WARM-UP SCHEMES

Our simulation protocol incorporates a "warm-up" period during which the reward is set to zero for the first several days so that learning begins only when the MSE estimates become more stable. We conduct ablations with warm-up lengths of 2, 5, 7, and 14 days using the real-data-based simulator with $(n, M) = (35, 12)$ and a $5\%$ policy lift. As reported in Table 2, shorter warm-up periods lead to noticeably higher MSE, indicating that noisy early-stage rewards degrade learning. The best performance is achieved with a 7-day warm-up (rather than 14), which strikes an effective balance between removing unstable early estimates and preserving sufficient training data.

Table 2: MSE ($\times 10^{-4}$) under Different Designs

| Design | BSZ | HW | XCT | NMDP | TMDP |
|---|---|---|---|---|---|
| MSE | 260.0 | 224.0 | 110.0 | 66.2 | 59.9 |
| **Design** | **WSY** | **TRL (warm-up=2)** | **TRL (warm-up=5)** | **TRL (warm-up=7)** | **TRL (warm-up=14)** |
| MSE | 46.5 | 42.6 | 40.1 | **37.9** | 39.2 |

Note: MSE ($\times 10^{-4}$) across 400 simulations. TRL with 7-day warm-up achieves the lowest MSE, confirming that sparse rewards (longer burn-in) outperform dense rewards (shorter burn-in). The optimal warm-up length is 7 days, not 14 days.

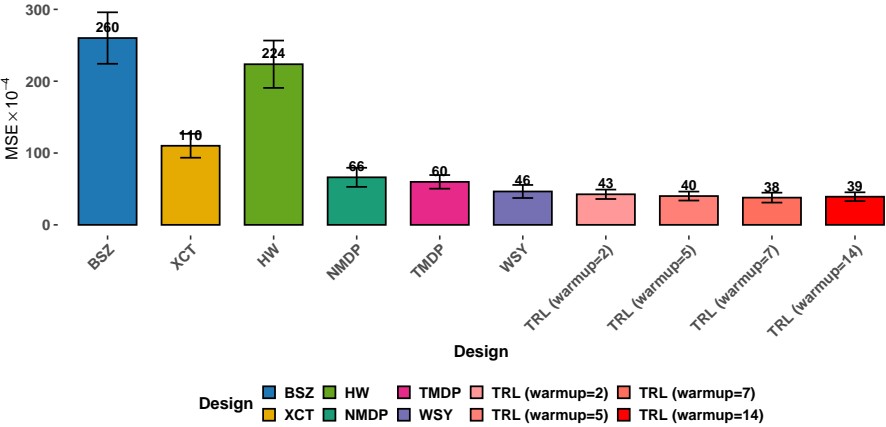

Figure 8: MSE ($\times 10^{-4}$) under different warm-up lengths. Shorter warm-up (2, 5 days) $\rightarrow$ higher MSE, validating sparse rewards.

### B.3 SENSITIVITY AND ROBUSTNESS OF SIMULATOR

To evaluate our algorithm's robustness to the misspecification of the simulator, we conducted a series of additional experiments based on Setting (i) from the Synthetic simulator in Section 4. Recall that without any misspecification, TRL achieves an MSE of 0.0027, whereas the best baseline, TMDP, has an MSE of 0.0036. We then introduced five types of shifts to misspecify the simulator:

- **Noise shift.** We increase the noise levels of equation 8, from $\sigma_y = \sigma_o = 0.2$ to $0.25$, resulting in an MSE of **0.0028**.

- **Covariate shift.** The initial covariate $O_{i,1}$ in equation 8 originally follow a bivariate standard normal distribution. We shift the mean to $0.1$ and $0.3$, obtaining MSEs of **0.0028** and **0.0030**, respectively.

- **Transition shift.** We modify a $2 \times 2$ transition matrix $\Phi_m$ in equation 8 by changing its last entry from $0.60$ to $0.55$, yielding an MSE of **0.0030**.

- **Effect-coefficient shift.** We increase the effect coefficient $\beta$ in equation 7 from $0.20$ to $0.25$, which gives an MSE of **0.0030**.

- **Reward shift.** We add a constant $+0.1$ to $Y_t$ in equation 7 to model a reward shift, resulting in an MSE of **0.0029**.

Across all five scenarios, TRL shows only a minor decline compared to the original setting without misspecification and consistently outperforms the best baseline. These results demonstrate that TRL maintains strong performance even when the simulator deviates moderately from the true environment.

### B.4 SENSITIVITY AND ROBUSTNESS OF HYPERPARAMETERS

The robustness of our method to its hyperparameters is crucial. We use Setting (i) from the Synthetic simulator with ($M = 4$, $n = 30$) as the base configuration and vary one hyperparameter at a time to create different settings. MSEs of estimators under our designs are reported in Table 3, which show that the performance of our method exhibits some fluctuation as the hyperparameters vary, but all results remain close to the MSE of **0.0027** reported in the main paper. At the same time, they are still clearly better than the best baseline, TMDP, whose MSE is **0.0036**. We conclude that our method's performance is stable across a range of key hyperparameter values, consistently outperforming the best baseline.

Table 3: Empirical MSEs for different hyperparameters.

| Hyperparameter | Value 1 | Value 2 (Main Paper) | Value 3 |
|---|---|---|---|
| Discount Factor ($\alpha$) | 0.0030 ($\alpha = 0.9$) | **0.0027** ($\alpha = 0.8$) | 0.0027 ($\alpha = 0.7$) |
| Transformer Width | 0.0027 (32) | **0.0027** (64) | 0.0030 (128) |
| Target Network Update Rate | 0.0027 (0.004) | **0.0027** (0.005) | 0.0029 (0.006) |
| Exploration Rate ($\epsilon$) | 0.0029 ($\epsilon = 0.03$) | **0.0027** ($\epsilon = 0.10$) | 0.0029 ($\epsilon = 0.15$) |
| **Best Baseline (TMDP)** | | **0.0036** | |

## B.5 Large time horizon

To evaluate the performance of proposed method with larger time horizons $T$, we conduct additional experiments in Setting (i) of the synthetic simulation. We fix $n = 30$ and let $M \in \{4, 8, 12, 16\}$, with $T = nM$.

Table 4 reports the results for different methods under larger values of $T$. As the time horizon increases, our method consistently outperforms all competing baselines, demonstrating its robustness to longer experimental horizons.

Table 4: Empirical MSE ($\times 10^{-4}$) across 400 replicates under different time horizons.

| Method | T=120 | T=240 | T=360 | T=480 |
|---|---|---|---|---|
| BSZ | 320 | 253 | 213 | 182 |
| XCT | 173 | 296 | 240 | 171 |
| HW | 98 | 317 | 256 | 173 |
| NMDP | 37 | 47.3 | 70.4 | 77.8 |
| TMDP | 36 | 45.3 | 70.1 | 77.9 |
| WSY | 39 | 55.7 | 64.7 | 71.2 |
| TRL | 27 | 41 | 55.2 | 69 |

## B.6 Computational complexity

We report the **empirical running time** in our experiments. In the synthetic simulator with setting (i), we fix $n = 30$ and vary $M \in \{4, 8, 12, 16\}$, and record the running time on an NVIDIA 2080 GPU. Under this configuration, the experiment with $n = 30$ and $M = 4$ finishes in a little over 3 hours, see Table 5. However, once the policy has been trained, generating an experimental design for a new A/B test is very fast: in our experiments, the deployment time is comparable to that of existing baselines and is typically less than one minute. Although our method is more time-consuming to train, it achieves higher estimation accuracy than the competing approaches, which can be highly valuable in practical applications.

Table 5: Training time (1000 epochs) under different horizons $T$.

| Horizon $T$ | Running Time(Hours) |
|---|---|
| 120 | 3:34:50.529 |
| 240 | 12:18:55.589 |
| 360 | 26:39:10.679 |
| 480 | 45:53:16.918 |

## B.7 Assessment of simulator availability in practice

When no simulator exists, we continue to use *setting (i)* in the *Synthetic* simulator and explore the sample (Days) needed before TRL meaningfully outperforms simple switchbacks, and the length. According to Table 6, we observe that when the burn-in length is around 9 days, the performance is comparable to switchback, and when it reaches 11 days, it clearly surpasses switchback. Hence, the required burn-in period 9 days is short relative to the duration of experiments 30 days.

Table 6: Effect of Different Burn-in Days on the Empirical MSEs

| Switchback | Day=5 | Day=7 | Day=9 | Day=11 |
|---|---|---|---|---|
| 0.003855 | 0.004419 | 0.00439 | **0.003899** | **0.003704** |

## B.8 EMPIRICAL CORRELATION ANALYSIS OF REAL DATA

To validate the high nonstationarity and long-horizon dependencies in the real data, we provide figures reporting the empirical autocorrelation and cross-correlation functions of demand and supply under the real-data–based simulator described in the main text, with n = 35, M = 12, and a 5% performance lift under the new policy.

We visualize the autocorrelation functions and cross-time correlations of gmv, the number of order requests, and total online time in Figures 9a and 9b, and the cross-correlations between order requests and drivers' total online time in Figure 9c. Figure 9a shows that all three variables exhibit strong positive lag-1 and lag-2 autocorrelations, followed by negative autocorrelations at lags 3–5. Figure 9b shows that order requests and drivers' total online time have non-zero, predominantly positive, cross-correlations. In addition, most cross-time correlations in Figure 9c are also positive. These dynamics violate the Markovian and stationarity assumptions underlying TMDP/NMDP, which struggle to capture such non-stationary, long-horizon dependencies. In contrast, TRL leverages full historical trajectories to adaptively model evolving relationships.

## B.9 SENSITIVITY EXPERIMENTS UNDER VARYING LEVELS OF CORRELATION

To validate the advantage of TRL under different correlation strengths, we firstly systematically rescale the demand–supply cross-correlation by factors in $\{0.25, 0.5, 1.0, 1.5\}$, with 1.0 corresponding to the main setting, by considering

$$\mathbb{E}[O_{m+1,1} \mid A_m, O_m] = \Phi_{m,11}O_{m,1} + (\phi_{\text{coef}}\Phi_{m,12})O_{m,2} + \Gamma_{m,1}A_m,$$

$$\mathbb{E}[O_{m+1,2} \mid A_m, O_m] = \Phi_{m,22}O_{m,2} + (\phi_{\text{coef}}\Phi_{m,21})O_{m,1} + \Gamma_{m,2}A_m.$$

TRL's relative MSE (vs. TMDP/NMDP) **decreases** as the correlation increases (Figure 10 and Table 7), indicating that its advantage grows precisely when temporal dependence is strongest and most non-stationary, thereby confirming that TRL excels in environments where historical context matters most.

Secondly, we conduct an experiment by adjusting the correlation strength among outcome residuals $\{\hat{e}_{i,m} : i = 1, \ldots, n, \; m = 1, \ldots, M\}$ in Algorithm 1. Let $\hat{\Sigma}_e = \frac{1}{n}\sum_{i=1}^{n}(\hat{\mathbf{e}}_i - \bar{\hat{\mathbf{e}}})(\hat{\mathbf{e}}_i - \bar{\hat{\mathbf{e}}})^{\top}$, $\hat{\mathbf{e}}_i = (\hat{e}_{i,1}, \ldots, \hat{e}_{i,M})^{\top}$, and write $\hat{\Sigma}_e = L_e L_e^{\top}$ with $D_e = \text{diag}(\hat{\Sigma}_e)$ and $R_{e,0} = D_e^{-1/2}\hat{\Sigma}_e D_e^{-1/2}$ denoting the empirical correlation matrix.

To vary dependence while keeping marginal variances fixed, we form a one-parameter family $\{R_e(\rho) : \rho \in [-1, 1]\}$ interpolating between independence, the empirical structure, and a more strongly correlated target $R_{e,1}$ (we take $(R_{e,1})_{mm'} = 0.6$ for $m \neq m'$). Specifically,

$$R_e(\rho) = \begin{cases} (1+\rho)\,R_{e,0} - \rho\,I_M, & \rho \in [-1, 0], \\ (1-\rho)\,R_{e,0} + \rho\,R_{e,1}, & \rho \in [0, 1], \end{cases}$$

so that $R_e(-1) = I_M$, $R_e(0) = R_{e,0}$, and $R_e(1) = R_{e,1}$. We then set $C_e(\rho) = D_e^{1/2}R_e(\rho)D_e^{1/2}$, $C_e(\rho) = L_e(\rho)L_e(\rho)^{\top}$, $A_e(\rho) = L_e(\rho)L_e^{-1}$, and define transformed residuals $\tilde{\mathbf{e}}_i(\rho) = A_e(\rho)\hat{\mathbf{e}}_i, i = 1, \ldots, n$. By construction, $\text{Var}(\tilde{\mathbf{e}}_i(\rho)) = C_e(\rho)$ and $\text{diag}C_e(\rho) = \text{diag}(\hat{\Sigma}_e)$ for all $\rho$. Thus, the marginal variances are preserved, while the temporal correlations are attenuated as $\rho \to -1$ and amplified as $\rho \to 1$. Replacing $\hat{\mathbf{e}}_i$ by $\tilde{\mathbf{e}}_i(\rho)$ in Algorithm 1 therefore yields a family of bootstrap simulators indexed by $\rho$, which we use to examine how the estimators' MSE varies with correlation strength. In our experiments, we consider $\rho \in \{-0.8, -0.4, 0, 0.4, 0.8\}$. Empirical results (Figure 11 and Table 8) reveal pronounced temporal persistence, strong cross-time correlations, and complex lead–lag structures—conditions under which TRL's full-history modeling is clearly advantageous compared with the Markovian assumptions underlying TMDP and NMDP.

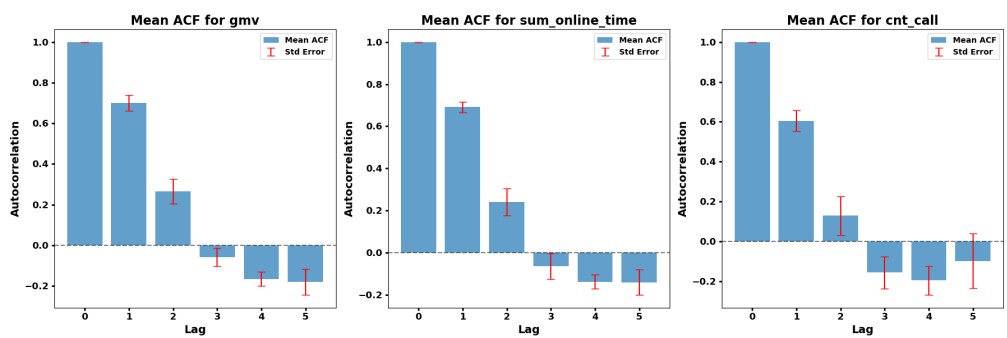

(a) Autocorrelation of *gmv* and state variables (lag 0 to 5).

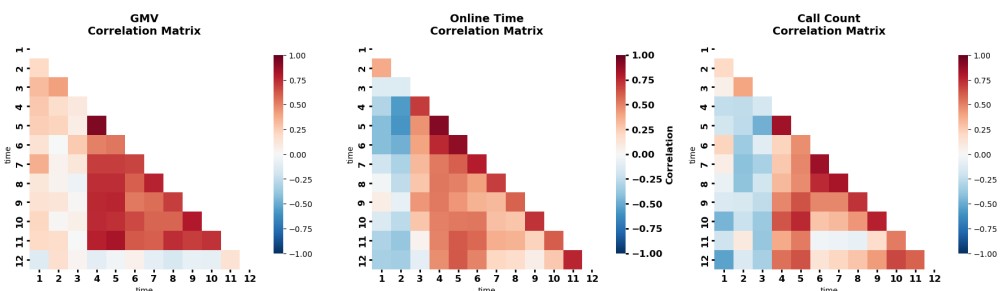

(b) Cross-time correlation matrices for *gmv*, *sum_online_time*, and *cnt_call*.

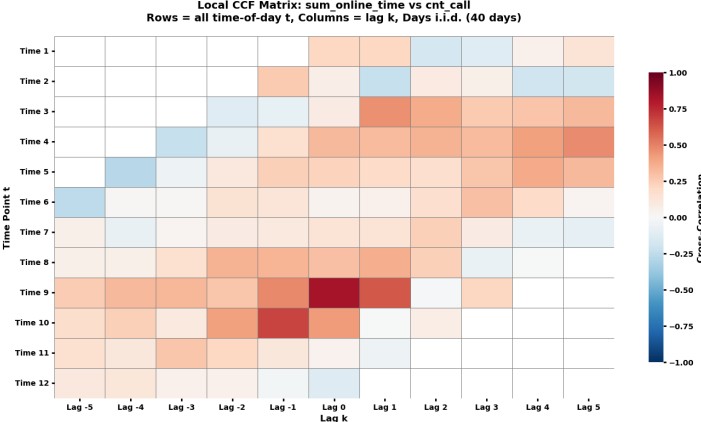

(c) Cross-correlation between demand (*sum_online_time*) and supply (*cnt_call*) across time points (t=1 to 12).

Figure 9: Empirical evidence of temporal and cross-series dependence in real data.

Table 7: MSE ($\times 10^{-4}$) by Method and Demand–Supply Correlation ($\phi_{coef}$)

| $\phi_{coef}$ | NMDP | TMDP | TRL |
|---|---|---|---|
| 0.25 | 58.4 | 56.5 | **31.6** |
| 0.50 | 60.3 | 56.9 | **32.3** |
| 1.00 | 66.2 | 59.9 | **39.2** |
| 1.50 | 75.8 | 66.4 | **40.3** |

Table 8: MSE ($\times 10^{-4}$) by Method and Residual Temporal Correlation ($\rho$)

| $\rho$ | NMDP | TMDP | TRL |
|---|---|---|---|
| -0.8 | 4.88 | 4.10 | **2.91** |
| -0.4 | 5.78 | 5.19 | **3.47** |
| 0.0 | 6.62 | 5.99 | **3.92** |
| 0.4 | 7.99 | 7.25 | **4.79** |
| 0.8 | 9.47 | 8.26 | **5.58** |

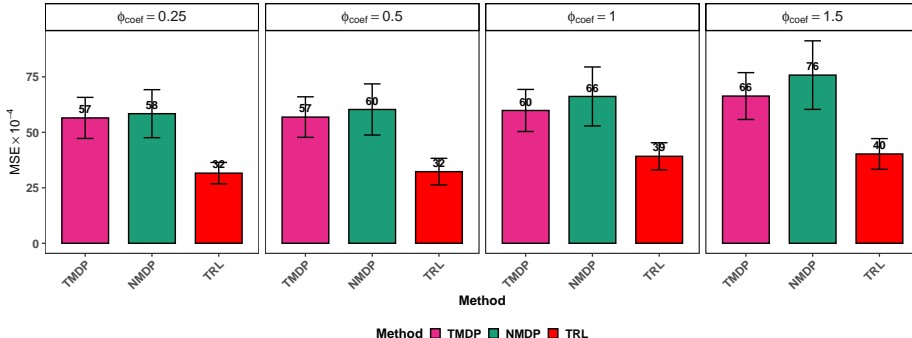

Figure 10: Empirical MSE ($\times 10^{-4}$) under different demand–supply correlation structures (parameterized by $\Phi$).

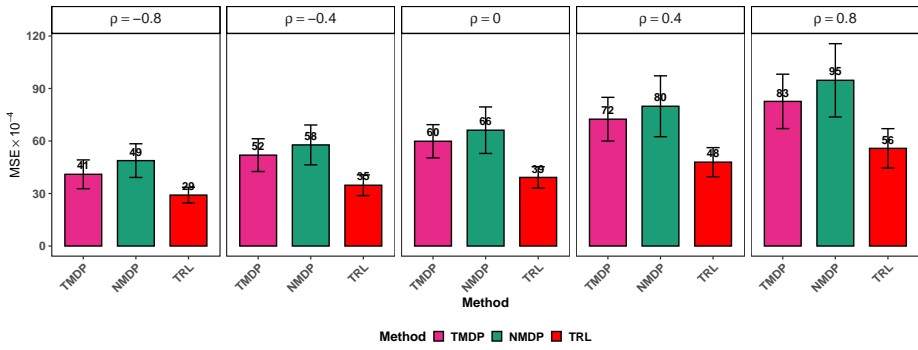

Figure 11: Empirical MSE ($\times 10^{-4}$) under Different Residual Correlation Levels (Parameterized by $\rho$).

