# OpenReview forum: "Designing Time Series Experiments in A/B Testing with Transformer Reinforcement Learning"
_ICLR.cc/2026/Conference — ICLR 2026 Poster_

### Official Review · Reviewer_U3U6 · 2025-10-23

**Soundness:** 1
**Presentation:** 3
**Contribution:** 2
**Rating:** 2
**Confidence:** 4

**Summary:**

This paper addresses the design of A/B experiments in time series settings, where existing methods often fail to use the full history or rely on restrictive modeling assumptions. The authors propose a new method, TRL, which models the experimental design as an RL problem. This method uses a Transformer to encode the entire past history as its state, and a double DQN agent learns a policy to select the next action (treatment or control). The agent is trained within a simulation environment to directly optimize the ATE's MSE by using a dense reward function based on the intermediate MSE. Empirical results on three different simulation environments show TRL outperforms existing designs.

**Strengths:**

1. The paper introduces a novel concept for a problem of practical importance. The idea of using modern sequence models and RL to directly optimize a statistical objective like MSE seems original.
2. The paper correctly identifies the limitations of traditional, more restrictive experimental designs. The problem is well motivated with clear context, and well situated within the literature.
3. The authors test against a wide range of relevant baselines in their empirical evaluation, demonstrating the strong performance of their method.

**Weaknesses:**

1. There seems to be a discrepancy between the paper’s core premise and the experimental setup. The paper is motivated by the need for a full history model to capture complex, non-markovian dynamics (as suggested by Theorem 1). However, in the appendix (if I understand correctly), the simulation environments appear to be Markovian. For example, the "Synthetic" and "Real data-based" simulators (Appendix A.2, Eq. 5, 6, 11) model the next state as a function of only the previous state and action. Could the authors clarify this? If the environments are indeed Markovian, it is not clear how the experiments are testing the paper's central hypothesis about the necessity of the full history.
2. The method's dependency on a simulator. The methodology seems to require a simulation environment to train in, and the reward function is defined by a "ground truth" ATE that must be pre-calculated from this simulator. This seems to suggest the method is for optimizing policies within a known simulator. Is this method applicable to real-world problems where a high-fidelity simulator is not available?
3. Usage of dense reward function: The paper notes that “early ATE estimators are less accurate due to smaller sample sizes." (in line 331, page 7). This raises a concern about training on a noisy signal and potential credit mis-assignment.
For example, if an action at an early step $t$ leads to a good-looking (but noisy) intermediate MSE purely by chance, the agent will be rewarded and may learn to prefer this action, even if it is not truly beneficial (or is even detrimental) to the final, stable MSE at time $T$. The agent may be learning a policy that optimizes for short-term statistical noise rather than the true, long-term objective. I think it could be helpful for the paper if the authors could provide an ablation study on the alpha parameter, particularly comparing the final design quality against the true, sparse-reward objective (where alpha=zero). This would validate that the dense reward is indeed a helpful heuristic and not a flawed proxy.

**Questions:**

Please address the weaknesses above.

In addition:
Regarding computational complexity: How is it possible to train the model in "approximately three GPU hours" when the method implies a quadratic complexity over sequences as long as $T=1008$ (24 hours x 42 days)? Please clarify the precise architecture (e.g., standard vs. linear Transformer) and provide a complexity analysis

I am open to changing the score if the authors can resolve these points.

---

> ### Author Response · Authors · 2025-11-21
> **Part 1**
>
> ## Reviewer U3U6
> Thank you for your thoughtful and critical assessment. Many of your comments will help us produce a more readable and self-contained version of the paper. Below, we address each of your specific concerns in turn.
>
> **The discrepancy between the paper’s core premise and the experimental setup (W1).**
>
> Thank you for this insightful comment. This point is indeed central to our work. You are correct that the simulation environments we use are Markovian in the sense that the data-generating process satisfies
>
> $$(Y_t, O_{t+1}) \perp H_{t-1} \mid (O_t, A_t)$$
>
> where each trajectory is $(O_t, A_t, Y_t),t=1, ..., T$, and $H_{t-1}$ denotes the history of observation–action–outcome triplets up to time $t-1$. This structure is indeed Markovian.
>
> However, the key quantity of interest in our paper is not the state-transition model itself, but the **mean squared error (MSE) of the ATE estimator**, which depends on the entire observed trajectory. Theorem 1 shows that even when the underlying environment is Markovian, the optimal experimental policy for minimizing the ATE estimator’s MSE is **inherently non-Markovian**: the optimal treatment assignment $A_t$ must depend on the full history, not only on $O_t$.
>
> Moreover, in our RL formulation, the reward is defined as the MSE of the current ATE estimator, which is itself a function of the full history. Therefore, the induced RL process
>
> $$(S_1, A_1, R_1, S_2, A_2, R_2, ...),$$
>
> where $S_t =(O_1, A_1, Y_1, \ldots, O_t)$, is **not Markovian**, even though the underlying environment $(O_t, A_t, Y_t)$  is Markovian.
>
> In summary, the experiments use Markovian simulators because the non-Markovian behavior arises not from the state-transition mechanism, but from (i) the structure of the statistical objective (MSE minimization) and (ii) the resulting optimal design policy—both of which necessarily depend on the full history, as predicted by Theorem 1.
>
>
> **The method's dependency on a simulator (W2).**
>
> Thank you for this insightful comment. Our TRL framework supports real-world deployment through a *sequential burn-in* mechanism, which can be viewed as a gradual, step-by-step way of improving the simulator using newly collected real-world data. After each batch of observations, the simulator is updated to better approximate the true environment, and the design policy is refined in an ongoing, adaptive manner. In this way, the design problem becomes a continuous optimization loop that grows more accurate as more data arrive. We also conducted additional experiments to illustrate the effectiveness of this sequential burn-in strategy; for details on the setup and results, please refer to our response to Weakness 3 of Reviewer RUcM.
>
> On the other hand, real-world platforms almost always maintain rich historical logs, for example from prior A/A tests, pilot regions, or analogous markets[1][2][3]. We can leverage these logs to build a data-driven simulator that enables deployment of our framework. This is particularly important for large-scale platforms such as Uber, Meituan, and Airbnb, where even small gains in experimental efficiency can lead to substantial business impact.
>
>
> **Usage of dense reward function (W3).**
>
> Thank you for this constructive comment. You are correct that the dense reward in Equation (3) may suffer from noisy intermediate signals, potentially leading to credit mis-assignment at early time steps. We address this from three perspectives:
>
> - **Flexibility in reward design.**
>   In practice, our reward is implemented in a much sparser form. In the simulations, we use the reward defined in Equation (9) of the Appendix:
>   $R_t = -\alpha^{n-i} \bigl[\widehat{\text{ATE}}(t) - \text{ATE}_{\mathrm{mc}}\bigr]^2$ if $t = iM$, and $R_t = 0$ otherwise.
>
>
>   where n is the total number of days and M is the number of time units per day. Thus, the MSE-based reward is updated only once per day using M         observations, and within-day rewards are zero. We presented a simplified dense reward in the main text for conceptual clarity, but the actual   implementation is already substantially sparser and less noisy. We will clarify this in the revision.
>
> - **Ablation on the discount parameter $\alpha$.**
>   We perform ablations with $\alpha$ $\in${0.7, 0.8, 0.9}. Details of the setup and results are given in our response to Weakness 5 of Reviewer RUcM.    As summarized in that Second Table, our method consistently outperforms the strongest baselines for all values of $\alpha$, indicating that the gains are not   tied to a specific dense-reward parameter.

---

> > ### Author Response · Authors · 2025-11-21
> > **Part 2**
> >
> > - **Warm-up to reduce early-stage noise.**
> >   Our simulation protocol includes a “warm-up” period, during which the reward is set to zero for the first several days so that learning starts only after the MSE estimates become more stable. We conduct ablations with warm-up lengths of 2, 5, 7, and 14 days using the real-data–based simulator with (n, M) = (35, 12) and a 5\% policy lift. As shown in Table X, shorter warm-up periods yield noticeably higher MSE, confirming that noisy early-stage rewards harm learning. The best performance is achieved with a 7-day warm-up (rather than 14), which balances removing unstable early estimates and retaining sufficient training data.
> >
> >
> >   **Table X. $MSE \times 10^{-4}$ under different designs.**
> >
> >   | Design                  | BSZ   | HW    | XCT   | NMDP  | TMDP  |
> >   |-------------------------|-------|-------|-------|-------|-------|
> >   | MSE                     | 260.0 | 224.0 | 110.0 | 66.2  | 59.9  |
> >
> >   | Design                      | WSY  | TRL (warm-up = 2) | TRL (warm-up = 5) | TRL (warm-up = 7) | TRL (warm-up = 14) |
> >   |-----------------------------|------|-------------------|-------------------|-------------------|--------------------|
> >   | MSE                         | 46.5 | 42.6              | 40.1              | **37.9**          | 39.2               |
> >
> > *Note:* MSE $\times 10^{-4}$ across 400 simulations. TRL with a 7-day warm-up achieves the lowest MSE, indicating that sparser rewards (longer burn-in) outperform denser rewards (shorter burn-in). The optimal warm-up length is 7 days rather than 14 days.
> >
> > **Clarify the precise architecture (e.g., standard vs. linear Transformer) and provide a complexity analysis (Q1)**
> >
> > We thank the reviewer for raising this question.
> >
> > - **Transformer architecture.**
> >   Across all experiments, we use a *standard* Transformer encoder rather than a linear-time variant. The model dimension is set to $\(d_{\text{model}} = 128\)$, meaning that all internal representations (input embeddings, attention outputs, feed-forward activations) lie in a 128-dimensional space. The encoder has two stacked layers, each consisting of a 4-head self-attention module and a position-wise feed-forward network, both with residual connections and layer normalization. Within each attention layer, the 128-dimensional hidden states are projected into four heads of dimension 32, allowing the model to capture complementary dependencies across time. After the encoder, we take the final hidden state as the contextual summary and feed it through a linear head to produce Q-values for all actions.
> >
> > - **Computational complexity.**
> >   A standard Transformer layer has $O(L^2)$ complexity in the sequence length L, which in our case corresponds to $O(T^2)$. To empirically confirm this scaling, we benchmark the runtime of our method on the synthetic simulator in setting (i), fixing $(n=30)$ and varying M $\in$ {4,8,12,16)
> > . Training is run on an NVIDIA 2080 GPU. For the smallest configuration $(n=30, M=4)$, full training takes just over 3 GPU hours (Table 1). By comparison, the largest configuration in the real-data–based simulator $(M=24,n=42)$, trained on an NVIDIA A100 GPU, requires 64.43 hours. These measurements are consistent with the quadratic scaling in \(L\), and the 3-hour runtime specifically corresponds to the smallest synthetic setting.
> >
> >
> >   **Table 1. Training time under different horizons $\(T = M \times n\)$.**
> >
> >   | Horizon $\(T = M \times n\)$ | Running Time (hours) |
> >   |----------------------------|----------------------|
> >   | 120                        | 3:34:50.529          |
> >   | 240                        | 12:18:55.589         |
> >   | 360                        | 26:39:10.679         |
> >   | 480                        | 45:53:16.918         |
> >
> > **References**
> >
> > - [1]Xu et al. (2018)Large-scale order dispatch in on-demand ride-hailing platforms: A learning and planning approach.KDD
> > - [2]Xiong et al. (2024)Data-driven switchback experiments: Theoretical tradeoffs and empirical bayes designs.Arxiv
> > - [3]Li et al.(2024) Combining Experimental and Historical Data for Policy Evaluation.ICML

---

> > > ### Comment · Reviewer_U3U6 · 2025-11-24
> > >
> > > I thank the authors for their detailed rebuttal. I have read the response carefully, as well as the reviews from the other reviewers.The rebuttal has successfully addressed my primary concerns, specifically regarding the validity of the experimental setup and the method's robustness.
> > >
> > > 1. Re W1: My initial review raised a concern that testing a "full-history" method in markovian simulation environments was a contradiction. The authors have provided a convincing theoretical justification: while the environment dynamics are markovian, the statistical objective (minimizing the MSE of the ATE estimator) creates a non-markovian optimization landscape. The optimal design must track historical assignments to minimize variance, even if the state transitions are memoryless. This resolves my concern regarding the validity of the experiments.
> > >
> > > 2. Re W3: I requested an ablation study to test if the dense reward function ($R_t$) introduced noise or credit mis-assignment. The authors provided this ("warm-up" ablation), showing that skipping early, noisy rewards (e.g., a 7-day warm-up) improves performance. This confirms the concern but effectively demonstrates that the method is robust when this hyperparameter is tuned.
> > >
> > > 3. Re Q1:The authors clarified that training for larger horizons takes approximately 64 GPU hours. While this confirms the high computational cost inherent in the quadratic complexity of transformers, I agree that this cost is acceptable for an offline experimental design phase, where the design is computed once prior to a multi-week experiment.
> > >
> > > In light of these clarifications, the solid empirical performance against relevant baselines, and the novelty of applying transformer-based RL to this domain, I have raised my score to 6.

---

> > > > ### Author Response · Authors · 2025-11-24
> > > >
> > > > Thank you for your detailed follow-up; we truly appreciate your acknowledgement of our efforts in addressing your concerns and for raising your score ! We will incorporate our response into the paper should it be accepted. Once again, we greatly appreciate your thoughtful and constructive feedback, which has  led to a significant improvement in the work.

---

### Official Review · Reviewer_hMwQ · 2025-10-26

**Soundness:** 3
**Presentation:** 2
**Contribution:** 2
**Rating:** 4
**Confidence:** 3

**Summary:**

The authors propose a transformer-based reinforcement learning (RL) approach for designing time-series experiments. In time-series experiments, the experiments should be conditioned on the entire history; the authors show that failing to do so yields suboptimal designs. However, conditioning on the entire history is challenging. Therefore, existing designs typically leverage only parts of the history. The algorithm in the paper then uses transformers to enable experiment design that depends on the entire history. They further use RL as an optimizer to avoid the simplifying assumptions commonly made in the literature. In synthetic and real-data experiments, the authors show that their approach outperforms various baselines.

**Strengths:**

1) The central contribution and gap in the literature are well presented.

2) The evaluation covers various baselines and both synthetic and real data.

3) The discussion of related literature is extensive.

**Weaknesses:**

1) While it is positive that the authors discuss a lot of related literature, the discussion itself is not as easy to follow. There are many references, but often little discussion to clarify the types of approaches presented. Also, the specific limitations are not always clear. For instance, when discussing A/B testing, it is said that some works relax the Markov assumption by modeling the data as a partially observable Markov decision process. Why is this still not enough to solve the problem?

2) The authors assume access to a simulation environment that approximates the data-generating process. What is then the motivation for using RL? One could also simply try out all possible designs. Or, if efficiency is a problem, use something like Bayesian optimization. RL is typically not very sample efficient and often gets stuck in local optima. So it would be good to argue why this is the "right" tool to use. Especially since, as an alternative to using a simulator, the authors propose collecting data throughout the day and then choosing a design for the next day. That is, there is a lot of time between experiments. Thus, a computationally more expensive optimizer could also be used.

3) I find Figure 1 not very self-explanatory, and both the caption and when it is mentioned in the text don't provide much explanation either.

4) The contribution itself to me feels rather limited. It seems like a rather straightforward application of RL + transformer to a specific problem.

5) Some smaller writing issues: I think the sentence part "identically distributed often leads to insignificant average treatment effect (ATE) estimator" doesn't work; the sentence part "motivated by applications in agricultural" is missing a word; the conditional distribution $\mathcal P_t$ should be properly defined. I would think about not using the word "treatment" in the abstract, as non-experts, who might still be interested in the paper, might lose interest as they could think this is related to medical applications.

**Questions:**

1) Can you motivate your choice of RL as an optimizer?

2) Is the carryover effect not mixing two things? It should be a problem that first, experiments are not iid, and second, that feedback is delayed.

3) Can you clarify the RL setup? Is it so that in each time step, the dimensionality of the state is growing?

4) How reliable is the physical simulator? Is there some uncertainty quantification, at least empirically?

5) What is the motivation for choosing exactly the mentioned baselines? Before, various lines of prior work have been discussed, so I think it would be good to motivate why now exactly those are selected as competitors.

---

> ### Author Response · Authors · 2025-11-21
> **Part 1**
>
> Thank you for your thoughtful and critical assessment. Many of your comments will help us produce a more readable and self-contained version of the paper. Below, we address each of your specific concerns in turn.
>
> **Literature Review and Distinction from Markov-Assumption-Relaxing Methods (W1).**
>
>   We have revised the discussion to clarify both the limitations of existing approaches and the distinct focus of our work.
>
> - First, we clarify that although some studies relax the Markov assumption by adopting POMDPs or more general time series models, these approaches still impose structural constraints on the data-generating process. Examples include finite-dimensional latent states, finite-memory carryover, or specific parametric dynamics. These constraints limit their ability to capture the long-range temporal dependencies that frequently arise in real A/B testing environments.
>
> - Second, our literature review in the A/B testing emphasizes ATE estimation because this is the central aim of most existing work. In contrast, our paper studies how to design and generate the experimental data itself. Even if more advanced estimation tools are available, a design that produces higher-quality data can still achieve superior overall performance. Our contribution introduces a conceptual shift by moving beyond the estimation framework and returning to the essence of experimental design. Unlike estimation-based methods that rely on specific modeling assumptions, our design procedure avoids such assumptions and enables end-to-end optimization of the MSE using RL.
>
> - Finally, we revise the literature review to more clearly distinguish between estimation and design, articulate the limitations of existing approaches, and more precisely motivate our methodological contribution.
>
> **The motivation of RL as an optimizer (W2\&Q1).**
>
>  We employ RL not as a generic optimizer, but because our experimental design problem is fundamentally a sequential decision process under full-history dependence, a setting for which RL is uniquely suited.
>
> - First, our goal is to minimize the final MSE of ATE estimators, which depends on the entire sequence of treatment assignments. At each time t, the design must select treatment A or B based on the full history $H_t$. Each action affects the next observation and ultimately affects the final MSE. This sequential, history-dependent structure is the core problem that RL is designed to solve, as it learns state-to-action mappings through Bellman backups and experience replay to efficiently assign credit across long horizons.
>
> - Second, exhaustive search is computationally impossible, even with a simulator. For a typical experiment with $n=20$ units and $T=24$ time steps, the number of possible treatment sequences is $2^{nT} \approx 10^{144}$, which is very large. No amount of offline computation or waiting time between days makes brute-force optimization viable.
>
> - Third, Bayesian optimization (BO) treats the simulator as a static black-box function $f(\theta)$ that maps a low-dimensional parameter vector (i.e. a fixed switchback interval or a small number of hyperparameters) to an objective value. This is fundamentally misaligned with our setting: BO optimizes static designs, not dynamic history-dependent policies. In addition, BO ignores non-Markovian temporal dependencies. Since the design must respond to long-range dependencies that affect carryover and MSE, a static black-box search over $\theta$ cannot capture the complex structure.
>
> In summary, we use RL because the design problem is itself a sequential, history-dependent decision problem, and RL is the natural computational framework for learning such dynamic policies. Together with the transformer architecture, it provides an assumption-free and efficient approach for adaptive experimental design.
>
> **The description of Figure 1 (W3).**
>
>  we have enhanced Figure 1 and its caption to make the end-to-end workflow of our TRL framework more intuitive and self-contained.
>
> - The updated figure now includes explicit annotations of the TRL workflow for adaptive A/B test design:
>    - The left panel shows the raw time-series history $(O_1,A_1,Y_1,\cdots,O_t)$ at time $t$, denoted as $S_t$, which serves as the input state for our RL policy.
>    -  A Double DQN agent uses $S_t$ to select the optimal treatment action $A_t$.
>    - The selected action is applied in a simulator, generating new data $(A_t, Y_t, O_{t+1})$, which updates the history for the next decision. The reward $R_t$ is computed as the MSE of the current ATE estimate.
>    - Both Q-networks in the Double DQN employ a Transformer architecture to summarize historical information.
>
> - We have added a clarifying sentence (page 2, lines 63-67):
>   “As illustrated in Figure 1, our TRL framework integrates a Transformer encoder to summarize the full historical context, which is then used by a Double DQN agent to output a dynamic treatment policy that minimizes the MSE of the ATE estimator.”

---

> ### Author Response · Authors · 2025-11-21
> **Part 2**
>
> **The contribution itself to me feels rather limited (W4).**
>
> Thank you for your feedback. We respectfully disagree and would like to clarify that our contribution is both substantial and novel, going far beyond a straightforward application of RL and Transformers. Our work makes three key contributions:
>
> 1. **A new problem formulation for A/B test design.**
>    To the best of our knowledge, we are the first to formulate A/B test design as a sequential decision problem under non-Markov dynamics. Prior work typically treats A/B test design as a static optimization over a few hyperparameters (e.g., switchback intervals [1]). In contrast, we cast the allocation strategy itself as a dynamic policy that must learn from the entire observed history to minimize a global objective (MSE). This represents a fundamental shift: from optimizing static designs to learning adaptive online policies that respond to historical data.
>
> 2. **A theoretically justified use of Transformers.**
>    The Transformer is not simply an architectural choice; it is essential for modeling full-history dependencies. Our impossibility theorem (Theorem 1) shows that full-history conditioning is necessary in this setting, thus making both our formulation and our Transformer-based solution novel in the A/B testing literature.
>
> 3. **A practical and deployment-ready framework.**
>    Our TRL framework is readily applicable in real-world settings because large-scale platforms routinely maintain rich historical logs collected from prior A/A tests, pilot deployments, and parallel markets[1][2][3]. These logs allow us to construct a data-driven simulator. This practical deployability is crucial for platforms such as Uber, Meituan, and Airbnb, where even modest gains in experimental efficiency can yield substantial business impact.
>
> In summary, although RL and Transformers are known components, our reformulation of A/B test design as an RL problem, together with the theoretical justification for full-history Transformer architectures, constitutes a distinct and meaningful contribution. We believe our framework advances the methodology of experimental design for time-series interventions in a significant way.
>
> **Some smaller writing issues (W5).**
>
> Thank you for your careful reading and helpful suggestions. We will revise the paper accordingly:
>
> - We will revise the sentence to:
>   *“If carryover effects are ignored and time-series observations are treated as independent and identically distributed, classical A/B testing methods can yield an estimated average treatment effect (ATE) that is statistically insignificant[4].”*
>
> - We will revise the phrase to:
>   *“motivated by applications in agriculture”*.
>
> - We will explicitly define the conditional distribution in  line 262.
>
> - We will revise the abstract by replacing “treatment” with “policies” to avoid unintended associations with medical applications.
>
> **Carryover effects vs. temporal dependence (Q2).**
>
> Thanks for raising this point. You are correct that the carryover effect seems to mix two aspects: (1) non-i.i.d. experiments, and (2) delayed feedback. In fact, following the literature such as [1][5], the carryover effect emphasizes that an action at time $t$ may influence both the current outcome $Y_t$ and future outcomes $Y_s$ for $s > t$. Consequently, some papers use “carryover effect” and “delayed effect” interchangeably [5]. While this does contribute to non-i.i.d. outcomes, it is not the only source of temporal dependence in A/B testing data. Therefore, in our paper, we interpret the carryover effect as the delayed effect, and we explicitly refer to temporal dependence when discussing the specific statistical implications.
>
> **The dimensionality of the state is growing (Q3).**
>
> We apologize for the confusion. You are correct that the state dimensionality grows over time. In our RL setup, the state at time $t$ is defined as the full history $S_t =(O_1, A_1, Y_1, \dots, O_t)$ as defined in line 334. This design is intentional, as our impossibility result (Theorem 1) demonstrates that conditioning on the full history is necessary in this setting.
>
> In practice, the growth remains manageable because A/B testing experiments in real companies are typically of limited duration due to cost constraints, as noted in point 3 of the challenges (lines 81-83). Specifically, these experiments often last only 3–5 weeks (21–35 days), with 12 or 24 time points per day. This results in a total of roughly 252–840 time points, which can be efficiently handled by Transformer-based architectures.
>
> If the sequence length becomes substantially larger, one could adopt ideas from online learning to construct low-dimensional summaries that capture all relevant information up to time $t$, similar to the hidden states in recurrent neural networks. This approach provides an effective dimensionality reduction of the state representation without changing the underlying TRL framework.

---

> > ### Author Response · Authors · 2025-11-21
> > **Part 3**
> >
> > **The analysis of some uncertainty quantification (Q4).**
> >
> > Thank you for this insightful comment. We address it in two parts:
> >
> > - We have conducted a series of experiments using a misspecified simulator to approximate the real environment. The results show that TRL maintains strong performance even when the simulator moderately deviates from the true environment.
> >
> > - Our TRL framework supports real-world deployment via a “sequential burn-in” mechanism, in which the simulator is continuously updated with live data. This effectively turns the design problem into an ongoing adaptive optimization loop. We also conducted simulations to demonstrate the performance of this sequential burn-in mechanism.
> >
> > For further details regarding the simulation setups and results, please refer to our response to Weakness 3 of Reviewer RUcM.
> >
> > **The motivation for choosing exactly the mentioned baselines (Q5).**
> >
> > In Section 2, we review several lines of prior work, including estimation-based approaches (Section 2.1) and experimental-design–based approaches (Section 2.2). The baselines we select correspond exactly to the most relevant methods summarized in item (iii) of Section 2.2 (lines 217–230), namely prior works on experimental design for A/B testing. These methods all rely on strong assumptions to make the MSE optimization problem tractable. For clarity, we group these assumptions into three categories:
> > (a) MDP-based assumptions[5][6][7][8];
> > (b) specific time-series model assumptions[1] [9][10];
> > (c) short carryover structures where delayed effects last only a few periods [11][12][13].
> >
> > For each category, we choose the most representative and state-of-the-art methods as baselines:
> >
> > - For (a), we include HW[7], TMDP/NMDP [8], and WSY[5], which are leading MDP-based design strategies.
> >
> > - For (b), we select XCT [1]. We do not include [9] or[10] because they rely on much more restrictive ARMA or Bayesian structural time-series assumptions, respectively. In contrast, XCT adopts substantially milder conditions, such as additive intervention effects (Condition 1 in [1]), which makes it the most appropriate and comparable baseline.
> >
> > - For (c), we choose BSZ[11], as the other two methods in this category are conceptually similar to BSZ.
> >
> > Overall, our baseline selection aims to cover all major assumption classes in the existing literature while focusing on the most representative and state-of-the-art methods in each class.
> >
> > **References**
> > - [1]Xiong et al. (2024)Data-driven switchback experiments: Theoretical tradeoffs and empirical bayes designs.Arxiv
> > - [2]Xu et al. (2018)Large-scale order dispatch in on-demand ride-hailing platforms: A learning and planning approach.KDD
> > - [3]Li et al.(2024) Combining Experimental and Historical Data for Policy Evaluation.ICML
> > - [4]Shi et al.(2023)Dynamic Causal Effects Evaluation in A/B Testing with a Reinforcement Learning Framework.JASA
> > - [5]Wen et al.(2025)Unraveling the Interplay between Carryover Effects and Reward Autocorrelations in Switchback Experiments.ICML
> > - [6]Glynn  et al.(2020) Adaptive experimental design with temporal interference: A maximum likelihood approach.NIPS
> > - [7]Hu et al. (2022)Switchback experiments under geometric mixing.Arxiv
> > - [8]Li et al(2024)Optimal treatment allocation for efficient policy evaluation in sequential decision making.NIPS
> > - [9]Sun et al. (2024)ARMA-Design: Optimal Treatment Allocation Strategies for A/B Testing in Partially Observable Time Series Experiments.Arxiv
> > - [10] Ni et al.(2025)Enhancing Efficiency and Robustness for Switchback Experiments: A Practical Model-assisted Framework.SSRN 5229804
> > - [11]Basse et al.(2023) Minimax designs for causal effects in temporal experiments with treatment habituation.Biometrika
> > - [12]Bojinov et al. (2023) Design and analysis of switchback experiments.Management Science
> > - [13]Chen et al.(2023)Efficient switchback experiments with surrogate variables: Estimation and experimental design. Management Science

---

> > > ### Comment · Reviewer_hMwQ · 2025-11-25
> > >
> > > I thank the authors for the clarifications and for addressing my concerns. I have adapted my score accordingly.

---

> > > > ### Author Response · Authors · 2025-11-26
> > > >
> > > > Dear Reviewer hMwQ,
> > > >
> > > > Thank you for your feedback. We are pleased to hear that our responses have addressed your concerns and that you have decided to raise the score. We will incorporate your valuable suggestions to improve the quality of our paper.
> > > >
> > > > We sincerely appreciate your support.

---

### Official Review · Reviewer_xX2C · 2025-10-26

**Soundness:** 3
**Presentation:** 2
**Contribution:** 3
**Rating:** 4
**Confidence:** 2

**Summary:**

The submitted manuscript studies the design of time series experiments in A/B testing where treatments are sequentially assigned and outmodes exhibit temporal dependencies. From a theoretical point of view, the authors derive an impossibility theorem showing that optimal design should consider full history. From a practical point of view they propose a transformer reinforcement learning approach. The proposed transformer architecture allows to allocate treatments on the entire history. Reinforcement learning is used to directly optimize the MSE. Empirical results on synthetic, simulated, and real ridesharing data indicate that TRL outperforms existing designs.

**Strengths:**

- The paper provides a comprehensive and well-structured literature review, clearly situating the work within the A/B testing, experimental design, and reinforcement learning communities.
- It proposes a novel use of reinforcement learning for experimental design, employing transformer architectures to condition treatment allocation on the entire observed history, encoded as an augmented state. This design choice is both technically interesting and conceptually well-motivated.
- The experimental evaluation is extensive, including multiple synthetic, simulated, and real-data environments with several relevant baseline methods. The empirical results appear to be convincing, showing consistent performance improvements of the proposed TRL approach over existing designs. Unfortunately, I cannot fully assess the correctness or completeness of all baselines due to limited domain familiarity.

**Weaknesses:**

My main concern lies in the formulation and proof of the main theoretical result (Theorem 1).

- The statement of Theorem 1 does not align with what is actually proved. The optimization problem (2) is formulated for an arbitrary ATE estimator, implying that the theorem should hold universally. However, the proof effectively fixes a specific estimator, and the argument depends crucially on that choice. As a result, the theorem as stated appears too general, and it is not demonstrated that the claim holds for all possible estimators. Moreover, it is not obvious that an optimal policy necessarily exists in the general formulation.
- In the proof, the authors fix a doubly robust ATE estimator and then derive a policy that minimizes the variance of this estimator. It is unclear, however, that minimizing the variance yields the same policy that minimizes the MSE, since the estimator may be biased. The potential dependence of the bias on the policy could alter the optimal solution, and the existence or uniqueness of such a minimizing policy is not established.

**Questions:**

Below are my questions and minor comments:
- How would the analysis or algorithm change if the ATE objective were replaced by a discounted or finite-sum objective over time?
- What are the assumptions on the ATE estimator used in Equation (2)? In particular, which assumptions are required for Theorem 1 to hold?
- The sentence „In other words, it is impossible for treatment allocation strategy in which $\pi_t$ omits dependence on any observation, action or outcome at any time step to be optimal.“ should be moved outside of Theorem 1, as it summarizes the result rather than constituting part of it.
- In the numerical experiments, the settings (i)–(iv) are not described in the main body. I had to refer to the appendix to find these details; including a brief description in the text would improve readability.
- How do the different methods used in the experiments compare in terms of running time? Finding an optimal policy in the augmented state-space formulation may be computationally challenging.
- Relatedly, how does the performance comparison change with larger time horizons T?
- The augmented state formulation seems memory-intensive, as it requires storing the entire history. Would it be possible to apply dimension reduction or use summarized state representations to mitigate storage costs?

---

> ### Author Response · Authors · 2025-11-21
> **Part 1**
>
> Thank you for your thoughtful and critical assessment. Many of your comments will help us produce a more readable and self-contained version of the paper. Below, we address each of your specific concerns in turn.
>
> **Restatement of Theorem 1 (W1\&Q2\&Q3).**
> For Theorem 1, we will list the required conditions and restate it for clarity.
>
>  - **Regular conditions.**
>     In the revision, we will explicitly state the following condition for the proof of Theorem 1.
>
>     Let $H_{t-1}$ denote the set of observation–action–outcome triplets up to time $t-1$. Consider the data generating distributions $(\mathcal{P}_t)_t$ that satisfy the following three conditions:
>
>   1. **Condition (CMIA) — Conditional mean independence assumption**
>
>      The conditional mean of $Y_t$ given $(O_t, A_t)$ is independent of $H_{t-1}$, for any t.
>
>   2. **Condition (CIA) — Conditional independence assumption**
>
>      The conditional distribution of $O_{t+1}$ given $H_t$ depends only on $(O_j)_{j \le t}$, for any t.
>
>   3. **Condition (HCVR) — History-dependent conditional variance ratio**
>
>      Let $\sigma_t^2(H_t, O_t, A_t)$ denote the conditional variance of $Y_t$ given $(O_t, A_t)$ and $H_t$. Then, for each t, $\sigma_t(H_t, O_t, A_t)$ is positive almost surely, and the ratio  $$\frac{\sigma_t(H_t, O_t, 1)}{\sigma_t(H_t, O_t, -1)}$$
>      depends on all variables in $H_t$ and $O_t$. That is, no subset of these variables is sufficient to fully recover this ratio.
>
>   **Theorem (Impossibility theorem).**
>     Suppose we set $\widehat{ATE}$ to the double robust estimator. Then there exist data generating processes $(\mathcal{P}_t\)_t$ under which the optimal policy $\pi$ that minimizes $\mathrm{Var}(\pi)$ depends on the entire past history for all $1 \le t \le T$, and this optimal policy is unique.
>
>
>
>
>
> **Impact of bias on the policy (W2）.**
> Theorem 1 is derived for the  double robust estimator class of estimators, such as the doubly robust estimator with correctly specified nuisance functions. In practice, however, correctly specifying these functions is often difficult and can introduce bias. Nevertheless, when the product of the estimation errors of the nuisance components $(\mu_t, \pi_t)$ converges faster than $o_p(n^{-1/2})$ (a standard condition in the semiparametric literature [1][2][3], the variance dominates the MSE. Within this class, there exists a unique doubly robust estimator that asymptotically minimizes (though does not exactly minimize) the MSE.
>
>
> **Discounted or finite-sum ATE objectives (Q1).**
> Our current ATE objective is already defined as a finite sum over time. If we replace it with a discounted version
>
> $$
> \text{ATE}
> = \frac{1}{T}\sum_{t=1}^T \kappa^{t}\bigl[ E_{1}(Y_t) - E_{-1}(Y_t) \bigr],
> $$
>
> our analysis and algorithm can be adapted with only minor changes.
>
> In settings with a linear transition model, we can follow the approach of [4] and estimate the discounted ATE via
>
> $$
> \hat{\text{ATE}} = \frac{2}{T}\sum_{t=1}^T \kappa^{t}\gamma_t + \frac{2}{T}\sum_{m=2}^T \kappa^{m}\beta_m^\top \left[ \sum_{k=1}^{m-1} \left(\Phi_{m-1}\Phi_{m-2}\cdots\Phi_{k+1}\right)\Gamma_k \right],
> $$
>
> where the weights $\kappa^{t}$ encode the discount factor. The Monte Carlo estimator $\text{ATE}_{\mathrm{MC}}$ can be modified analogously by simulating trajectories under the model and applying the same discount weights.
>
> For more general model-free settings, we can first estimate the potential outcome processes using standard off-policy evaluation methods (e.g., doubly robust estimation or marginal sensitivity approaches such as [4][5][6][7], and then obtain the discounted effect by applying fitted Q-learning or value-function estimation to the two potential-outcome MDPs. The difference between the resulting policy values corresponds to the discounted $\widehat{\text{ATE}}$.
>
> **The brief description of the experiments (Q4).**
>
>   Thank you for this helpful suggestion. Due to space constraints, we did not provide full descriptions of Settings~(i)–(iv) in the main text. Details of    the settings are present in Appendix A.2.

---

> > ### Author Response · Authors · 2025-11-21
> > **Part 2**
> >
> > **Running Time of the Experiments (Q5).**
> > We report the empirical running time in our experiments. In the synthetic simulator with setting (i), we fix n= 30 and vary  M $\in$ \{4, 8, 12, 16\}, and record the training time on an NVIDIA 2080 GPU. Under this configuration, the experiment with n = 30 and M = 4 finishes in a little over 3 hours (see Table 1). However, once the policy has been trained, generating an experimental design for a new A/B test is very fast: in our experiments, the deployment time is comparable to that of existing baselines and is typically less than one minute.Although our method is more time-consuming to train, it achieves higher estimation accuracy than the competing approaches, which can be highly valuable in practical applications.
> >
> > **Table 1. Training time  under different horizons T.**
> >
> > | Horizon \(T\) | Running Time (hours) |
> > | ------------- | -------------------- |
> > | 120  | 3:34:50.529         |
> > | 240  | 12:18:55.589        |
> > | 360 | 26:39:10.679        |
> > | 480 | 45:53:16.918        |
> >
> >
> > **Additional experiments with larger time horizons T (Q6).**
> >
> > We conduct additional experiments in Setting (i) of the synthetic simulation.We fix n = 30, M $\in$ {4,8,12,16\},with $T = nM$.
> > Table 2 reports the results for different methods under larger values of T. As the time horizon increases, our method consistently outperforms all competing baselines, demonstrating its robustness to longer experimental horizons.
> >
> > **Table 2. Empirical means of MSE ($\times 10^{-4} $) across 400 replicates under different time horizons.**
> >
> > | Method | T = 120 | T = 240 | T = 360 | T = 480 |
> > |--------|-------------|-------------|-------------|-------------|
> > | BSZ    | 320         | 253         | 213         | 182         |
> > | XCT    | 173         | 296         | 240         | 171         |
> > | HW     | 98          | 317         | 256         | 173         |
> > | NMDP   | 37          | 47.3        | 70.4        | 77.8        |
> > | TMDP   | 36          | 45.3        | 70.1        | 77.9        |
> > | WSY    | 39          | 55.7        | 64.7        | 71.2        |
> > | TRL    | 27          | 41          | 55.2        | 69          |
> >
> >
> > **Memory cost of the augmented state formulation (Q7).**
> >  We agree that the space complexity of the augmented state is an important practical concern. In our intended deployment scenarios, however, the effective horizon is relatively modest: the number of experimental days n is typically 3–5 weeks, each day has only 12 or 24 time intervals. Combined with a 7-dimensional feature vector, the total sequence length and embedding size remain small compared to standard NLP models (e.g., 512-768 dimensions), and the memory footprint is well within the capacity of modern hardware.
> >
> > If the sequence length were to become substantially larger, one could borrow ideas from online learning and construct low-dimensional summary statistics that summarize all information up to time t, rather than storing the raw full history. This would effectively reduce the dimensionality of the state representation without changing the TRL framework[8][9][10].
> >
> > **References**
> > - [1]Chernozhukov et.al(2018)Double/debiased machine learning for treatment and structural parameters.Econometrics Journal
> > - [2]Kallus et.al(2020)Double reinforcement learning for efficient off-policy evaluation in markov decision processes.JMLR
> > - [3]Tsiatis et.al(2007) Semiparametric Theory and Missing Data.Springer Science
> > - [4]Luo et.al(2024)Policy evaluation for temporal and/or spatial dependent experiments.JRSSB
> > - [5]Shi et.al(2022)Statistical inference of the value function for reinforcement learning in infinite-horizon settings. JRSSB
> > - [6]Shi et.al(2023) Dynamic Causal Effects Evaluation in A/B Testing with a Reinforcement Learning Framework.JASA
> > - [7]Uehara et.al(2022)A review of off-policy evaluation in reinforcement learning.Arxiv
> > - [8]Wang et.al(2020)Linformer: Self-attention with linear complexity.Arxiv
> > - [9]Choromanski et.(2020)alRethinking attention with performers.Arxiv
> > - [10]Chen et.al(2021)Skyformer: Remodel Self-Attention with Gaussian Kernel and Nystrom Method.NIPS

---

> ### Comment · Reviewer_xX2C · 2025-11-27
> **Thank you for your response**
>
> Dear Authors,
>
> Thank you for the thoughtful response, the clarifications, and the additional experiments. I appreciate the effort to revise the statement of assumptions. That said, my evaluation from a theoretical perspective remains largely unchanged. In its current form, I find the statement and communication of the main theoretical contribution still misleading relative to what is actually proved. In short, I see two separate mismatches: one between the stated contribution and the formal theorem, and another between the theorem as stated and the arguments provided in its proof.
>
> On page 2 you write:
> > “Specifically, we establish an impossibility theorem to prove that it is generally impossible for allocation strategies that do not fully leverage the entire history to achieve optimality.”
>
> Theorem 1 does not establish this general claim. More precisely:
> 1. The result (as acknowledged in your response) is proved for a particular estimator, not for “allocation strategies” in general. As stated in the contribution paragraph, the claim reads as an estimator-agnostic impossibility theorem; the theorem is not estimator-agnostic.
> 2. The proof optimizes variance of the chosen estimator, while the theorem's objective is MSE. For the (generally biased) estimator under consideration, minimizing variance need not minimize MSE. The asymptotic variance-dominance discussion for certain $(\mu_t,\pi_t)$ does not fix this: the asymptotics themselves depend on $\pi_t$ (the policy being optimized), there is no argument that the required asymptotics hold for the optimal policy, and the theorem does not claim asymptotic MSE optimality. As a result, the stated “impossibility” for MSE-optimality is not established.
> 3. Even setting bias aside, the theorem and its proof do not establish uniqueness of the optimal policy, nor that every optimal policy must depend on the entire path history. Multiple optimal policies can exist; some may not condition on all past observations/actions at some times $t$.
>
> Given the remaining concerns, I will keep my score unchanged.

---

> > ### Author Response · Authors · 2025-12-02
> > **Reviewer xX2C (Follow-up questions)**
> >
> > We thank the reviewer for the constructive follow-up comments. In the revised paper, we have addressed the two mismatches raised by the reviewer (see our revised statement of contributions, Theorem 1, and its proof, where major changes are highlighted in red). Below we provide a concise summary of our changes:
> >
> > 1. In the stated contributions, we now clearly specify that our result is established for the restricted class of doubly robust estimators. We also clarify that the guarantees are asymptotic (Page 2 line 101).
> >
> > 2. Prior to stating Theorem 1, we reiterate that our analysis focuses on this class of doubly robust estimators and note that minimizing their MSE is asymptotically equivalent to minimizing their asymptotic variance (Page 6 lines 295–297).
> >
> > 3. Theorem 1 has been revised to state that the optimal policy minimizes the asymptotic variance, rather than the finite-sample MSE (Page 6 lines 290–292).
> >
> > 4. The proof has been updated to explicitly establish the uniqueness of the optimal policy (Page 19 lines 1000–1024).
> >
> > We note that the first three bullet points address the mismatch between the stated contributions and Theorem 1, while the last two address the mismatch between the theorem and its proof.

---

### Official Review · Reviewer_RUcM · 2025-10-31

**Soundness:** 2
**Presentation:** 3
**Contribution:** 2
**Rating:** 4
**Confidence:** 3

**Summary:**

The paper tackles A/B testing when treatments are applied over time—like switching policies on a ride-sharing platform—and outcomes today depend on what happened earlier (carryover effects). It first proves a negative result: any design that doesn’t look at the entire past history when deciding the next treatment can be sub-optimal for minimizing the mean-squared error (MSE) of the final ATE estimate. Building on that, the authors propose a Transformer Reinforcement Learning (TRL) design strategy: use a transformer to condition treatment choices on the full history, and use RL to directly optimize a proxy for the ATE’s MSE. The approach is trained via a transformer-based DDQN with standard deep-RL tricks. Empirically, TRL yields lower MSE ATE estimates than switchback and MDP-based baselines.

**Strengths:**

Importance. I like that the paper targets time-series A/B testing where carryover effects are the norm; it seems very relevant for real platforms that roll out policies sequentially.

Empirical breadth. The breadth of experiments (synthetic, real-data-based, public simulator) and the many replications with CIs looks good; it seems the comparisons are careful.

The overall presentation of the paper is clear.

**Weaknesses:**

Scope and assumptions of Theorem 1. The impossibility result hinges on constructing settings where optimal assignment depends on the full history; could you (a) delimit the regularity conditions under which this dependence is strictly necessary and (b) discuss practically checkable conditions indicating when shorter-memory designs are near-optimal? (Right now, the proof shows existence rather than prevalence.)

You define ATE and optimize MSE, but multiple ATE estimators appear (e.g., OLS in linear cases, LSTD in the nonlinear dispatch setting); can you clarify the exact estimator used in each environment,  and whether the reward’s squared error is an unbiased (or consistent) proxy for the final-time MSE across those choices?

Since the reward uses a simulator-based ATEmc as “ground truth,” how sensitive are results to simulator misspecification, and what happens if the simulator embeds the same linearity assumptions used to generate training data (risking optimistic bias)? Could you add stress tests that deliberately mis-specify transition or reward models?

Availability of simulators in practice. You note the design can iterate day-by-day when no simulator exists; could you quantify the sample (days) needed before TRL meaningfully outperforms simple switchbacks, and provide guidelines for when the sequential bootstrap is stable enough to guide next-day designs?

Discounting and hyperparameters. The reward uses a discount α and your DDQN uses several training choices; can you include ablations on α, transformer depth/width, context length, target-network update rate, and exploration ε, showing how final ATE-MSE responds? Note that this is an important ablation study.

Since your point is “full-history helps,” could you compare TRL to (a) GRU/LSTM-DDQN with identical training budgets, and (b) transformer variants with restricted attention windows, to isolate gains from long-range attention vs. generic sequence modeling?

You attribute TMDP/NMDP under-performance in the A/A-based simulator to positive outcome autocorrelation; could you report empirical autocorrelation functions and cross-correlations (demand/supply) and show how performance degrades as you dial this correlation up/down?

**Questions:**

see above.

---

> ### Author Response · Authors · 2025-11-21
> **Part 1**
>
> Thank you for your thoughtful and critical assessment. Many of your comments will help us produce a more readable and self-contained version of the paper. Below, we address each of your specific concerns in turn.
> - **Scope and assumptions of Theorem 1 (W1).**
>   Theorem 1 applies to settings with non-Markovian temporal dependence, where the conditional outcome variance depends on the entire past history (lines 965-966). It establishes that there exist plausible time-series environments in which any design that omits part of the history is strictly suboptimal, thereby ruling out the existence of a universal short-memory design class that remains optimal across all environments.
>
>   - **Regular conditions.**
>      We explicitly state the following conditions for the proof of Theorem 1.
>
>      Consider the data generating distributions $(\mathcal{P}_t)_t$ that satisfy the following three conditions:
>
>   1. **Condition (CMIA) — Conditional mean independence assumption**
>
>      The conditional mean of $Y_t$ given $(O_t, A_t)$ is independent of $H_{t-1}$, for any t.
>
>   2. **Condition (CIA) — Conditional independence assumption**
>
>      The conditional distribution of $O_{t+1}$ given $H_t$ depends only on $(O_j)_{j \le t}$, for any t.
>
>   3. **Condition (HCVR) — History-dependent conditional variance ratio**
>
>      Let $\sigma_t^2(H_t, O_t, A_t)$ denote the conditional variance of $Y_t$ given $(O_t, A_t)$ and $H_t$. Then, for each t, $\sigma_t(H_t, O_t, A_t)$ is positive almost surely, and the ratio  $$\frac{\sigma_t(H_t, O_t, 1)}{\sigma_t(H_t, O_t, -1)}$$
>      depends on all variables in $H_t$ and $O_t$. That is, no subset of these variables is sufficient to fully recover this ratio.
>
>   - **Near-optimality of short-memory design.**
>       The optimality of short-memory designs is rarely studied in the existing literature. Although a few works address this problem—such as the ARMA-based design of [1] and the finite–carryover-order optimal switchback design of [2]—these approaches rely on strong and specific assumptions, such as an ARMA model structure or prior knowledge of the order of carryover effects. In practical applications, however, one must first test whether such assumptions hold, for example through goodness-of-fit tests for ARMA models [3] [4] or procedures for determining the carryover order [2] [5].
> - **ATE estimator and consistent proxy for MSE (W2).**
>     We adopt different outcome models in different environments and clarify this in the revision.
>
>   - Specifically, in the Synthetic and Real data-based simulators, the ATE is estimated by an OLS estimator. In the Public dispatch simulator, rewards are generated by a fixed MDP. For each policy, we estimate its long-run average reward using LSTD method.
>
>   - When the data-driven simulator accurately captures the real environment, $ATE_{mc}$ is close to the ground-truth ATE, so the squared reward error becomes an essentially unbiased proxy for the final-time MSE across design choices. When there is a mismatch between the real environment and the data-driven simulator, this proxy becomes biased. However, our ultimate goal is to find an optimal design that minimizes the MSE of the ATE estimator in the true environment. In **the misspecification and robustness experiments**, we show that even when such a mismatch exists, the optimal design returned by TRL still delivers strong performance.
>  - **Robustness to the misspecification of the simulator (W3).**
>       To evaluate our algorithm’s robustness to the misspecification of the simulator, we conducted additional experiments based on Setting (i) from the Synthetic simulator. Without any misspecification, TRL achieves an MSE of 0.0027, whereas the best baseline, TMDP, has an MSE of 0.0036. We used five types of shifts to  misspecify the simulator:
>
>    - **Noise shift.** We increase the noise levels of equations 7, 8 in the 1038-1040 lines, from $\sigma_o=\sigma_y=0.2$ to $0.25$, resulting in an MSE of **0.0028**.
>
>    - **Covariate shift.** The covariates $O_{i,1}$ in equation 8 originally follow a bivariate standard normal distribution. We shift the mean to $0.1$ and $0.3$, obtaining MSEs of **0.0028** and **0.0030**, respectively.
>
>    - **Transition shift.** We modify a $2\times 2$ transition matrix $\Phi_m$ in equation 8 by changing its last entry from $0.60$ to $0.55$, yielding an MSE of **0.0030**.
>
>    - **Effect-coefficient shift.** We increase the effect coefficient $\beta_m$ in equation 7 from $(0.60,0.20)$ to $(0.65,0.25)$, which gives an MSE of **0.0030**.
>
>    - **Reward shift.** We add a constant $+0.1$ to $Y_t$ in equation 7 to model a reward shift, resulting in an MSE of **0.0029**.
>
>   Across all five scenarios, TRL shows only a minor decline compared to the original setting without misspecification and consistently outperforms the best baseline. These results demonstrate that TRL maintains strong performance even when the simulator deviates moderately from the true environment.

---

> > ### Author Response · Authors · 2025-11-21
> > **Part 2**
> >
> > - **Availability of simulators (W4).**
> >   This is another excellent comment. Below, we discuss two approaches for constructing the simulator: one based on historical data and another that works even without historical data.
> >
> >   - **Leveraging historical observational data.**
> >     Real-world platforms almost always maintain rich historical logs—for instance, from prior A/A tests, pilot regions, or analogous markets[6] [7] [8]. We can leverage these logs to build a data-driven simulator that allows us to quantify the sample-size requirements and stability thresholds you are concerned about.
> >
> >   - **Iterative day-by-day procedure.**
> >     In the absence of sufficient historical data (e.g., in a new market or for a new product), the experiment itself can be used to build the simulator in a iterative manner. We first run a short burn-in phase under a simple, safe design (e.g., uniform random assignment or a conservative switchback). Each day, the newly collected data are used to update the estimated data-generating process and hence the simulator, and the updated simulator is then plugged into the TRL framework to compute an improved design for the next day. This loop—updating both the simulator and the design—repeats as more data become available.
> >
> > - **Samples needed before TRL meaningfully outperforms simple switchbacks (W4).**
> > When no simulator exists, we continue to use setting (i) in the Synthetic simulator and explore the sample (Days) needed before TRL meaningfully outperforms simple switchbacks, and the length. According  to the following Table, we observe that when the burn-in length is around 9 days, the performance is comparable to switchback, and when it reaches 11 days, it clearly surpasses switchback.
> >
> >     | Switchback | Day=5   | Day=7   | Day=9        | Day=11        |
> >     |-----------|---------|---------|--------------|---------------|
> >     | 0.003855  | 0.004419| 0.004390| **0.003899** | **0.003704**  |
> > - **The Sensitivity analysis of hyperparameters (W5).**
> >   The robustness of our method to its hyperparameters is crucial. We have included additional experiments to illustrate how our method performs under different hyperparameter configurations.
> >
> >   The table below reports empirical means of MSEs for Setting (i) from the Synthetic simulator ($M=4$, $n=30$) while varying one hyperparameter at a time. The results show that the performance of our method exhibits some fluctuation as the hyperparameters vary, but all results remain close to the MSE of **0.0027** reported in the main paper. At the same time, they are still clearly better than the best baseline, TMDP, whose MSE is **0.0036**. We conclude that our method's performance is stable across a range of key hyperparameter values, consistently outperforming the best baseline.
> >
> >   | **Hyperparameter**              | **Value 1**                     | **Value 2 (Main Paper)**                 | **Value 3**                     |
> >   |---------------------------------|---------------------------------|------------------------------------------|---------------------------------|
> >   | Discount Factor ($\alpha$)      | 0.0030 ($\alpha=0.9$)          | **0.0027** ($\alpha=0.8$)                | 0.0027 ($\alpha=0.7$)          |
> >   | Transformer Width               | 0.0027 (32)                    | **0.0027** (64)                          | 0.0030 (128)                   |
> >   | Target Network Update Rate      | 0.0027 (0.004)                 | **0.0027** (0.005)                       | 0.0029 (0.006)                 |
> >   | Exploration Rate ($\epsilon$)   | 0.0029 ($\epsilon=0.03$)       | **0.0027** ($\epsilon=0.10$)             | 0.0029 ($\epsilon=0.15$)       |
> >   | **Best Baseline (TMDP)**        | **0.0036**                     |                                          |                                |
> > - **Ablation study on LSTM Models with different attention window sizes (W6).**
> >   Thanks for the insightful comment. We conduct an additional experiment in Setting (i) to compare our full-history Transformer (TRL) against two key ablations: (1) an LSTM-based DDQN with an identical training budget, and (2) Transformer variants with restricted attention windows.
> >
> >   The table below shows that sequence modeling based on LSTM already offers a substantial advantage over traditional methods such as TMDP (MSE: 0.0036). Our full-history Transformer (TRL) further improves upon LSTM, achieving a lower MSE. Moreover, we observe that, for the Transformer variants, increasing the attention window size consistently leads to better performance, indicating the benefit of exploiting longer-range temporal dependencies.
> >
> >   | TRL        | LSTM      | Window size 6 | Window size 12 | Window size 24 |
> >   |-----------|-----------|---------------|----------------|----------------|
> >   | **0.002750** | 0.002975  | 0.003002      | 0.002915       | 0.02871        |

---

> > > ### Author Response · Authors · 2025-11-21
> > > **Part 3**
> > >
> > > - **Performance across different temporal dependences (W7).**
> > >   We detail the temporal dependencies in the dataset,  conduct additional experiments to evaluate TMDP/NMDP's performance during the rebuttal and report the results below.  All the figures and details can be found in subsections B.8 and B.9 in the updated paper.
> > >
> > >   - **Temporal dependences in real data.**
> > >       We visualize the autocorrelation functions and cross-time correlations of gmv, the number of order requests, and total online time in Figures 9(a) and 9(b), and the cross-correlations between order requests and drivers’ total online time in Figure 9(c). Figure 9(a) shows that all three variables exhibit strong positive lag-1 and lag-2 autocorrelations, followed by negative autocorrelations at lags 3–5. Figure 9(b) shows that order requests and drivers’ total online time have non-zero, predominantly positive, cross-correlations
> > >   In addition, most cross-time correlations in Figure 9(c) are also positive.
> > >
> > >   - **Effects of cross- and auto-correlation on TMDP/NMDP.**
> > >     We conduct additional experiments varying strengths of demand–supply cross-correlation under the real data-based simulator from the main text with $n=35$, $M=12$, and a 5% performance lift under the new policy.
> > >     (i) To examine the effect of cross-correlation, we introduce a factor $\phi_{\text{coef}}$ in the simulator to control the magnitude of the correlation between order requests and drivers’ total online time, where a larger $\phi_{\text{coef}}$ indicates stronger cross-correlation. The below table shows that NMDP and TMDP perform poorly when $\phi_{\text{coef}} = 1.5$, corresponding to strong positive correlation. As the correlation weakens, their MSEs decrease. However, the proposed method consistently outperforms both across all scenarios.
> > >
> > >
> > >       | $\phi_{\text{coef}}$ | NMDP  | TMDP  | TRL         |
> > >       |----------------------|-------|-------|-------------|
> > >       | 0.25                 | 58.4  | 56.5  | **31.6**    |
> > >       | 0.50                 | 60.3  | 56.9  | **32.3**    |
> > >       | 1.00                 | 66.2  | 59.9  | **39.2**    |
> > >       | 1.50                 | 75.8  | 66.4  | **40.3**    |
> > >
> > >     (ii) For the auto-correlation experiment, we modify the residual covariance matrix in the $\textit{gmv}$ model to preserve marginal variances while scaling cross-time correlations by a factor $\phi \in$ {-0.8, -0.4, 0, 0.4, 0.8}, where larger $\phi$ indicates stronger auto-correlation. The following table shows that the MSEs of NMDP and TMDP increase with $\phi$, whereas TRL consistently achieves the lowest MSE across all settings.
> > >
> > >     | $\phi$ | NMDP | TMDP | TRL   |
> > >     |--------|------|------|-------|
> > >     | -0.8   | 4.88 | 4.10 | **2.91** |
> > >     | -0.4   | 5.78 | 5.19 | **3.47** |
> > >     | 0.0    | 6.62 | 5.99 | **3.92** |
> > >     | 0.4    | 7.99 | 7.25 | **4.79** |
> > >     | 0.8    | 9.47 | 8.26 | **5.58** |
> > >
> > > **References**
> > > - [1]Sun et al. (2024)ARMA-Design: Optimal Treatment Allocation Strategies for A/B Testing in Partially Observable Time Series Experiments.Arxiv
> > > - [2]Bojinov et al. (2023) Design and analysis of switchback experiments.Management Science
> > > - [3]Hallin, M., & Puri, M. L. (1988). Optimal rank-based procedures for time series analysis: Testing an ARMA model against other ARMA models. The Annals of Statistics.
> > > - [4]Francq, C., Roy, R., & Zakoïan, J.-M. (2003). Goodness-of-fit tests for ARMA models with uncorrelated errors. Working paper.
> > > - [5]Chen and Hong.(2012)Testing for the Markov property in time series.Econometric Theory
> > > - [6]Xu et al. (2018)Large-scale order dispatch in on-demand ride-hailing platforms: A learning and planning approach.KDD
> > > - [7]Xiong et al. (2024)Data-driven switchback experiments: Theoretical tradeoffs and empirical bayes designs.Arxiv
> > > - [8]Li et al.(2024) Combining Experimental and Historical Data for Policy Evaluation.ICML

---

> > > > ### Comment · Reviewer_RUcM · 2025-11-27
> > > >
> > > > Thank you for the response and the additional experiments. They addressed most of my concerns. I raised the score.

---

> > > > > ### Author Response · Authors · 2025-11-28
> > > > >
> > > > > Dear Reviewer RUcM,
> > > > >
> > > > > Thank you very much for raising the score. We will carefully incorporate your valuable suggestions to further improve the quality of our paper.
> > > > >
> > > > > We sincerely appreciate your support.

---

### Author Response · Authors · 2025-12-02
**Summary of responses**

Dear Area Chair,

We sincerely thank reviewers and the AC for their time, effort, and constructive comments. We provide a summary of our responses for your convenience when making your decision.

Before doing so, we would like to note that our rebuttal led to productive discussions with the reviewers, and three reviewers raised their scores to 6. These updates, along with the reviewers’ explanations, remain visible on OpenReview. Additionally, several of these changes occurred earlier in the discussion timeline, which reflect the reviewers’ scientific judgment following our clarifications.

### 1. Theory

Both Reviewer RUcM (Weakness 1) and Reviewer xX2C (Weakness 1, 2; Questions 1, 2) raised concerns about the mismatch between the scope and assumptions of our theoretical results and presentation. We have implemented the following clarifications:

 1. We now explicitly state in both the contributions and Theorem 1 that the result applies to the *doubly robust estimators* and is                *asymptotic*.

 2. We summarize standard results for this class of estimators, making clear that, under mild regularity conditions, the *asymptotic MSE equals the variance*. This addresses the mismatch between our presentation and proof.

 3. We explicitly list the *conditions on the data-generating process* used in the proof and we *prove the uniqueness of the optimal policy*, which was previously implicit.

For detailed discussions, please refer to:

- Rebuttal Part 1 to Reviewer RUcM (“Scope and assumptions of Theorem 1”).
- Rebuttal Part 1 to Reviewer xX2C (“Restatement of Theorem 1; Impact of bias on the policy”).

### 2. Experiments

We addressed all reviewer concerns regarding the empirical evaluation from four perspectives, and we have incorporated all new experimental results into Appendix Section B:

1. Reviewer RUcM (Weakness 3), Reviewer hMwQ (Question 4), and Reviewer U3U6 (Weakness 2) were concerned about the impact of simulator misspecification.     We conducted extensive experiments under five types of misspecification: noise, covariate, transition, effect-coefficient, and reward shifts. Across      all settings, our method exhibits strong robustness. Detailed results appear in *Rebuttal Part 1 to Reviewer RUcM (Robustness to the misspecification     of the simulator)*. In addition, Reviewer RUcM (Weakness 4) required us to quantify how many samples (days) are required before TRL meaningfully          outperforms simple switchbacks in the absence of a simulator. The corresponding results are provided in *Rebuttal Part 2 to Reviewer RUcM (Samples        needed before TRL meaningfully outperforms simple switchbacks)*.

3. Reviewer RUcM (Weakness 5) and Reviewer U3U6 (Weakness 3) requested ablations on key hyperparameters. We performed systematic analyses on the             attention window, discount factor, transformer width, target-network update rate, and exploration rate. Our method consistently outperforms the           strongest baseline across all variants. Full results are provided in *Rebuttal Part 2 to Reviewer RUcM (Sensitivity analysis of hyperparameters)*.

4. Reviewer xX2C (Questions 5, 6) and Reviewer U3U6 (Question 1) raised concerns about scalability as $T$ increases. We conducted dedicated experiments      evaluating both performance and runtime for larger $T$, and the method scales well in both aspects. Details are given in *Rebuttal Part 2 to Reviewer     xX2C*.

5. Reviewer RUcM requested two additional comparisons:  (i) LSTM models with different attention-window sizes to illustrate history dependence (Weekness 6), and (ii) analyses of the impact of cross-correlation on performance (Weekness 7). Both comparisons were implemented, with experimental setups and results presented in *Rebuttal Part 2 and Part 3 to Reviewer RUcM*.

### 3. Presentation

We improved the structure and writing of the revised main text, where definitions, intermediate steps, and key messages were clarified. In particular, following Reviewer hMwQ’s suggestions, we (i) added explanations of the specific limitations in Section 2.1 (Weakness 1), (ii) expanded the caption of Figure 1 to clarify its interpretation (Weakness 3), and (iii) included the definition and discussion of $\mathcal{P}_{t}$ in the problem setup in Section 3 (Weakness 5). We also strengthened the motivation and positioning in the Introduction and Related Work, and elaborated on potentially confusing concepts (see *the red-highlighted portions in the revised manuscript*).


Since we revised the main text, some line numbers in the previous rebuttal have changed, and we have accordingly updated the corresponding lines in the rebuttal. Once again, we greatly appreciate the reviewers’ insightful comments and timely engagement during the rebuttal. Thank you for your time and for considering our submission.

Sincerely,
The Authors

---

### Meta-Review · Area_Chair_yjNg · 2026-01-05

**Summary:**

The reviewers agree the paper tackles an important and practical problem: designing time series A/B experiments when there’s temporal dependence. Framing this as a sequential decision problem is seen as a strong and original idea. Using transformer-based RL to condition on full history is well motivated, and the impossibility result showing short-memory designs can fail adds weight. The experiments are extensive—synthetic data, simulators, and a real-world ridesharing dataset—and consistently show clear gains over existing designs. Some concerns were raised about the scope of the theory, reliance on simulators, and whether the contribution goes beyond applying RL and transformers, but the rebuttal addressed most of these with clarifications, revised statements, and many extra experiments. Several reviewers raised their scores after discussion. Overall, the paper is considered solid and timely, opening a new direction for adaptive experimental design, and meets the bar for acceptance.

**Reviewer Concerns:**

The rebuttal clarified the scope and assumptions of the impossibility theorem, restated it in asymptotic terms, added robustness experiments under simulator misspecification, and provided extensive ablations. Presentation and motivation were improved. Remaining concerns about generality and reliance on RL and simulators are now clearly acknowledged as limitations, not flaws.

**Reviewer Scores:**

Most reviewers who were marginal or slightly negative increased their scores after the rebuttal. One remained skeptical about the theory but did not oppose acceptance. Overall, the discussion led to a clear upward shift and consensus toward acceptance.

---

### Decision · Program_Chairs · 2026-01-26

Accept (Poster)